# Show, Don't Tell: Morphing Latent Reasoning into Image Generation

Harold Haodong Chen [* 1 2]   Xinxiang Yin [* 1]   Wen-Jie Shu [3]   Hongfei Zhang [1]   Zixin Zhang [1 4]
Chenfei Liao [1]   Litao Guo [1]   Qifeng Chen [† 2]   Ying-Cong Chen [† 1 2]

## Abstract

Text-to-image (T2I) generation has achieved remarkable progress, yet existing methods often lack the ability to dynamically reason and refine during generation–a hallmark of human creativity. Current reasoning-augmented paradigms mostly rely on explicit thought processes, where intermediate reasoning is decoded into discrete text at fixed steps with frequent image decoding and re-encoding, leading to inefficiencies, information loss, and cognitive mismatches. To bridge this gap, we introduce `LatentMorph`, a novel framework that seamlessly integrates implicit latent reasoning into the T2I generation process. At its core, `LatentMorph` introduces four lightweight components: (*i*) a **condenser** for summarizing intermediate generation states into compact visual memory, (*ii*) a **translator** for converting latent thoughts into actionable guidance, (*iii*) a **shaper** for dynamically steering next image token predictions, and (*iv*) an RL-trained **invoker** for adaptively determining when to invoke reasoning. By performing reasoning entirely in continuous latent spaces, `LatentMorph` avoids the bottlenecks of explicit reasoning and enables more adaptive self-refinement. Extensive experiments demonstrate that `LatentMorph` **(I)** enhances the base model Janus-Pro by $16\%$ on GenEval and $25\%$ on T2I-CompBench; **(II)** outperforms explicit paradigms (*e.g.*, TwiG) by $15\%$ and $11\%$ on abstract reasoning tasks like WISE and IPV-Txt, **(III)** while reducing inference time by $44\%$ and token consumption by $51\%$; and **(IV)** exhibits $71\%$ cognitive alignment with human intuition on reasoning invocation. Our code: `LatentMorph`.

## 1. Introduction

Text-to-image (T2I) generation has progressed rapidly in recent years, driven by advances in diffusion (Rombach

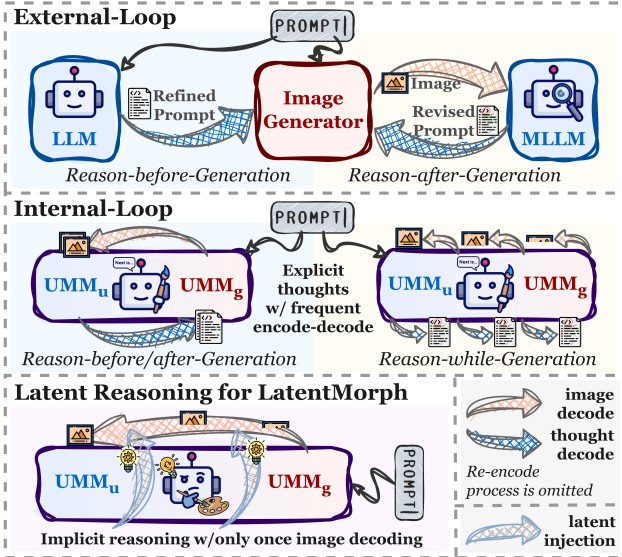

*Figure 1.* The comparison of reasoning-augmented image generation paradigms: external-loop, internal-loop, and `LatentMorph`.

et al., 2022; Saharia et al., 2022) and autoregressive (Sun et al., 2024; Wang et al., 2024a) generative models, which now power a wide range of applications. Despite these successes, previous T2I generators primarily function as text-to-pixel mapping systems, with limited capacity for explicit deliberation or self-refinement during generation–key hallmarks of human creativity.

In contrast, large language models (LLMs) (Brown et al., 2020; Touvron et al., 2023) have demonstrated remarkable emergent reasoning abilities, popularized by chain-of-thought (CoT) (Wei et al., 2022) prompting and training. This has motivated a growing line of work aiming to endow image generation with "System-2"-like reasoning. Existing approaches can be broadly organized into two paradigms: **(I) external-loop paradigm**, which couples an (M)LLM with a generator and uses the (M)LLM as an optimizer to refine prompts, critique outputs, or iteratively propose edits (Wang et al., 2024b; Gani et al., 2024; Yang et al., 2024; Zhan et al., 2024); and **(II) internal-loop paradigm**, increasingly enabled by unified multimodal models (UMMs) (*e.g.*, Janus-series (Chen et al., 2025c)), attempt to interleave reasoning between the understanding and generation branches within a single backbone (Liao et al., 2025; Huang et al., 2025b; Qin et al., 2025a; Guo et al., 2025b). While these reasoning-before, reasoning-after, or reasoning-while-

---
[*]Equal contribution  [1]HKUST(GZ) [2]HKUST [3]ZODA [4]Knowin. Correspondence to: Qifeng Chen <cqf@ust.hk>, Ying-Cong Chen <yingcongchen@usk.hk>.

*Proceedings of the 43$^{rd}$ International Conference on Machine Learning*, Seoul, South Korea. PMLR 306, 2026. Copyright 2026 by the author(s).

generating show effectiveness, a common thread remains: they typically rely on decoupled, explicit CoT, where "thinking" is forced to be decoded into discrete text and then re-ingested *at predefined fixed steps*.

This reliance on explicit thought introduces three fundamental deficiencies: (*i*) **information loss**: forcing intermediate cognition into natural language compresses rich internal states into a narrow symbolic channel (Zhu et al., 2025b); (*ii*) **inefficiency**: repeated decode-encode cycles add latency and consume context budget; (*iii*) **cognitive mismatch**: human creativity rarely involves verbalizing intermediate judgments at every step; instead, humans rely on continuous, implicit thoughts to guide their actions dynamically. Empirical validations are demonstrated in Section §5.

Given these limitations, **latent reasoning** offers a compelling alternative, where intermediate reasoning is performed in continuous hidden states rather than explicit text. Representative works in language modeling show that latent reasoning can be induced by special tokens that suppress explicit thought (*e.g.*, Coconut (Hao et al., 2024)), or by compressing verbose rationales into a small set of informative latent vectors (*e.g.*, SoftCoT (Xu et al., 2025)). While these approaches have been extended to multimodal reasoning tasks, such as visual latent thoughts for understanding (Li et al., 2025a; Dong et al., 2025), they remain constrained to a single understanding feature space. Directly applying these methods to image generation is non-trivial, as the task involves a distinct execution process with its own tokenization and dynamics. This introduces a bi-directional interface mismatch: a *perception gap*, where generation-time states are not directly interpretable by the understanding branch, and an *execution gap*, where reasoning-time states are not directly actionable for the generation branch. This leads to our pivotal research question:

> 🔍 *How can we seamlessly interleave latent reasoning into the image generation, enabling the model to dynamically monitor its evolving visual state and intervene to refine generation?*

To bridge this gap, we introduce LatentMorph, a dynamic and cognitively-aligned framework that morphs on-the-fly latent reasoning into T2I generation, eliminating the need for explicit textual CoT. At its core, LatentMorph closes the loop between reasoning and generation by introducing four lightweight components: First, a ♣ **condenser** that summarizes intermediate generation states into compact, semantically meaningful visual memory, enabling the understanding branch to process visual evidence efficiently without requiring prohibitively large state readouts. Second, a ♦ **translator** that then converts latent thoughts from the understanding branch into generator-compatible guidance, and a ♥ **shaper** that integrates these signals to steer image-token predictions dynamically. Additionally, an RL-trained

♠ **invoker** that adaptively decides when to invoke reasoning as a cognitive monitor, enabling a more cognitively-aligned process and overcoming the inefficiencies of fixed-step explicit reasoning injections in existing paradigms.

By performing implicit, interleaved reasoning during generation, LatentMorph enables adaptive self-refinement, improving instruction-following and compositional fidelity while avoiding the bottlenecks of explicit thoughts and the burden of constructing additional training data. Moreover, LatentMorph is model-agnostic and can be instantiated in both external-loop and internal-loop settings with autoregressive generators, with further analysis in Appendix §A.3.

**Empirical Evaluation.** Extensive experiments across five benchmarks and ten baseline strategies demonstrate that LatentMorph achieves ❶ **superior fidelity**, enhancing the base model Janus-Pro by 16.0% on GenEval and 25.3% on T2I-CompBench; ❷ **abstract reasoning**, outperforming the explicit reason-while-generation paradigm by 15.6% and 11.3% on the complex WISE and IPV-Txt benchmarks; ❸ **inference efficiency**, surpassing baselines while reducing inference time and token consumption by 44.3% and 51.0%, respectively; and ❹ **cognitive alignment**, where LatentMorph's adaptive invocation strategy achieves a 71.8% alignment with human evaluators.

## 2. Related Work

**Reasoning in Image Generation.** A growing line of T2I work seeks to incorporate LLM-style reasoning (Wei et al., 2022) to enable more cognitively aligned generation. Existing approaches largely fall into two families: **(I) external-loop** approaches couple an (M)LLM with a generator, most commonly as reasoning-before-generation (Wang et al., 2024b; Gani et al., 2024), *e.g.*, using an LLM for planning or prompt optimization, or reasoning-after-generation (Yang et al., 2024; Zhan et al., 2024), *e.g.*, adopting an MLLM for verification and revision via re-prompting or editing. In contrast, **(II) internal-loop** approaches, enabled by recent UMMs (Chen et al., 2025c; Xie et al., 2024; Team, 2024; Deng et al., 2025), interleave reasoning between the understanding and generation branches within a single backbone; beyond reasoning-before/after (Liao et al., 2025; Huang et al., 2025b; Qin et al., 2025a; Mi et al., 2025), recent work explores reasoning-while-generating (Guo et al., 2025b) by inserting periodic checks or interventions at predefined decoding steps. While effective, these paradigms most rely on decoupled, explicit reasoning, which causes information loss, or, like recent works adapting LatentSeek (Li et al., 2025b) to T2I (Mi et al., 2025), resort to test-time latent search. The latter also necessitates iterative reason-before/after interactions guided by external rewards, remaining computationally prohibitive and cognitively disjointed.

**Latent Reasoning in LLMs.** With LLM reasoning increasingly studied, recent work has shifted from explicit

CoT to latent reasoning, motivated by the representational richness and potential efficiency of latent spaces (Zhu et al., 2025b; Chen et al., 2025d). The core idea is to carry intermediate deliberation in compact latent representations rather than verbose textual rationales (Wang et al., 2023; Goyal et al., 2023; Deng et al., 2023). Recent works broadly follow two directions: **(I) special thinking-token** based implicit reasoning (Hao et al., 2024; Li et al., 2025a; Dong et al., 2025; Yang et al., 2025; Qin et al., 2025b) introduces dedicated tokens or control mechanisms to induce internal deliberation while suppressing explicit rationales, typically trained with supervision from textual reasoning traces or auxiliary multimodal cues; and **(II) distillation/compression** approaches (Xu et al., 2025; Shen et al., 2025; Zhang et al., 2025b;a) explicitly compress long CoT rationales into a small set of informative soft tokens that can be consumed more efficiently at inference. Although the aforementioned works mainly focus on the understanding setting within a single feature space, they have strongly inspired our exploration of latent deliberation for visual generation.

## 3. Preliminary: Problem Formalization

We formalize reasoning-augmented T2I generation in the context of a UMM, which consists of an autoregressive T2I generation branch $\text{UMM}_g$ and a multimodal understanding branch $\text{UMM}_u$. Given a user prompt $T$, $\text{UMM}_g$ autoregressively generates a sequence of discrete image tokens $X = (x_1, x_2, ..., x_{|X|})$, where $|X|$ is the total number of tokens. The probability of $X$ is modeled as:

$$p_\theta(X \mid T) = \prod_{i=1}^{|X|} p_\theta(x_i \mid T, x_{<i}), \qquad (1)$$

where $x_{<i} = (x_1, \ldots, x_{i-1})$ represents the tokens generated up to step $i - 1$. The sequence $X$ is decoded into the final image $\hat{I} = \text{Dec}(X)$. To ensure that the generated image $\hat{I}$ aligns with $T$, the objective is to maximize a reward function $R(\hat{I}, T)$ over a prompt distribution $\mathcal{D}$, defined as:

$$\max_\theta \mathbb{E}_{T \sim \mathcal{D}} \left[ R(\hat{I}, T) \right], \qquad (2)$$

where $R(\cdot)$ measures alignment metrics, *e.g.*, semantic alignment. To achieve this goal, reasoning interventions are introduced to leverage $\text{UMM}_u$'s reasoning capabilities for refining the generation process. At an intervention step $k \in \{0, 1, ..., |X|\}$, the partially generated tokens $X_{<k}$ are decoded into an intermediate image $\hat{I}_{<k} = \text{Dec}(X_{<k})$. This intermediate image, along with $T$, is encoded and passed to $\text{UMM}_u$ as $\text{Enc}(\hat{I}_{<k}, T)$. $\text{UMM}_u$ performs reasoning and generates intermediate thoughts $S_k = (s_1, s_2, ..., s_m, T')$, where $s_1, s_2, ..., s_m$ denote reasoning steps and $T'$ represents the refined prompt. $T'$ is then re-encoded as $\text{Enc}(T')$ and passed back to $\text{UMM}_g$ to continue generation.

For the reason-before/after-generation (*e.g.*, IRG (Huang et al., 2025b) and Uni-CoT (Qin et al., 2025a)), reasoning is invoked only at $k = 0/|X|$, respectively, and $T = T'$ for all subsequent steps (*e.g.*, generate from scratch), while for the reason-while-generation (*e.g.*, TwiG (Guo et al., 2025b)), reasoning is performed at fixed intermediate steps ($k \in \{k_1, k_2, ...\}$), *e.g.*, every $1/3 \, |X|$ generated tokens. The external-loop paradigm (*e.g.*, Idea2Img (Yang et al., 2024)) replaces $\text{UMM}_u$ and $\text{UMM}_g$ with separate (M)LLMs and image generators. Our work, which enables reasoning ($S_k$) entirely in the latent space, eliminates the need for decoding intermediate steps into text or images and re-encoding, along with designing a dynamic intervention mechanism that adaptively determines when ($k$) to reason, ensuring seamless integration with the generation process.

## 4. Methodology

### 4.1. LatentMorph: Generating with Latent Reasoning

Just as humans dynamically reflect and refine their thoughts while creating art, image generation can benefit from reasoning. However, existing methods rely on explicit reasoning, where intermediate thoughts are decoded into discrete text or images at fixed steps. To bridge this gap, LatentMorph interleaves latent reasoning directly into the generation process, enabling implicit and adaptive self-refinement.

As shown in Figure 2, given a user prompt $T$, the autoregressive generation branch $\text{UMM}_g$ generates image tokens $X = (x_1, x_2, ..., x_{|X|})$, where the evolving hidden states $\mathbf{H}_{1:i} = (\mathbf{h}_1, \ldots, \mathbf{h}_i)$ are monitored continuously. LatentMorph evaluates whether reasoning is required and integrates latent thoughts into the generation process when necessary. Specifically, like how humans continuously think while drawing, at every window interval $w$, a **short-term condenser** $\mathcal{C}_{\text{short}}$ compresses the most recent $\mathbf{H}_{i-w:i}$ into a local, short-term latent memory:

$$\mathbf{M}^{(s)} = \mathcal{C}_{\text{short}}(\mathbf{h}_{i-w}, \ldots, \mathbf{h}_i), \quad \mathbf{M}^{(s)} \in \mathbb{R}^{n_s \times d}, \qquad (3)$$

which does not incur excessive inference delay, as validated in Section §5.4. A **reasoning invoker** $\mathcal{I}_{\text{invoker}}$ then evaluates the current generation state $s_i$, which includes $\mathbf{M}^{(s)}$ and additional features (*e.g.*, prediction uncertainty), to determine whether reasoning should be invoked:

$$\pi_\theta(a_k \mid s_i) = \sigma(\mathcal{I}_{\text{invoker}}(s_i)), \qquad (4)$$

where $a_k \in \{\texttt{CONTINUE}, \texttt{REASON}\}$. If $a_k = [\texttt{CONTINUE}]$, $\text{UMM}_g$ continues generating tokens. Otherwise ($a_k = [\texttt{REASON}]$), the latent reasoning is activated at the intervention point $k = i$. Specifically, a **long-term condenser** $\mathcal{C}_{\text{long}}$[1] first summarizes the entire generation history $\mathbf{H}_{1:k}$ into a concise long-term visual latent memory $\mathbf{M}^{(l)} \in \mathbb{R}^{n_l \times d}$ by $\mathcal{C}_{\text{long}}(\mathbf{h}_1, \ldots, \mathbf{h}_k)$. This memory, along with the encoded prompt $T$, is passed to the reasoning branch $\text{UMM}_u$, without decoding and re-encoding itself, which performs latent reasoning and outputs latent thoughts

---

[1] We denote both short-term condenser and long-term condenser as the condenser in Section §1 for clarity.

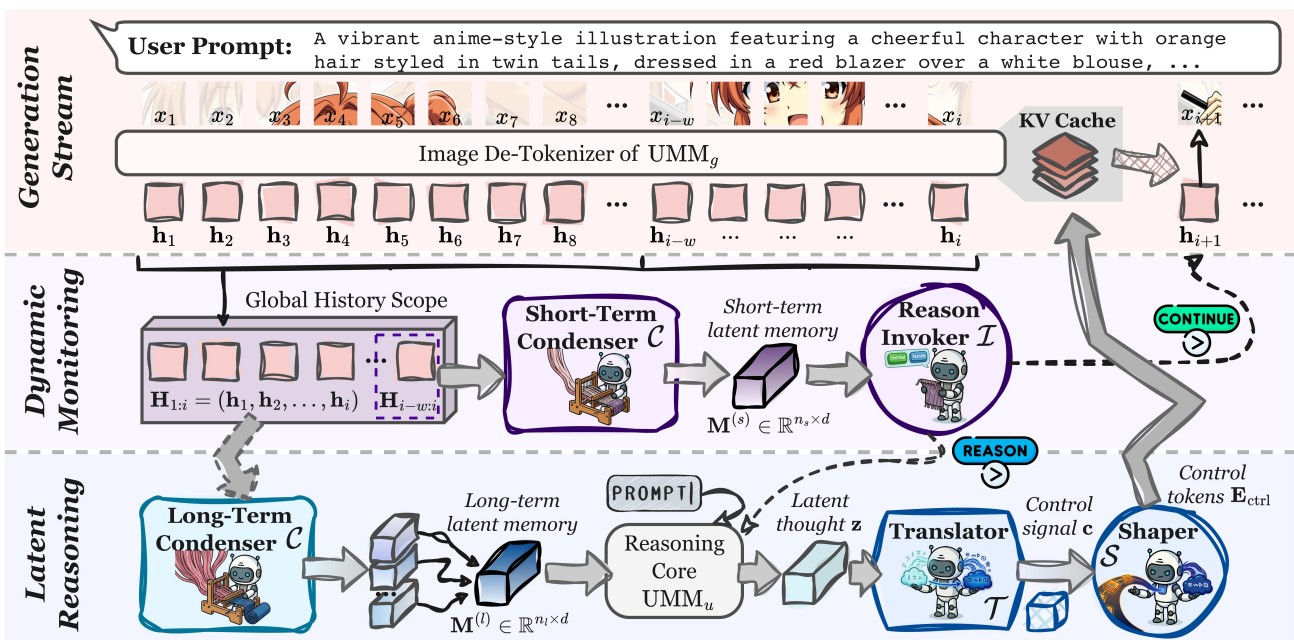

*Figure 2.* Overview of `LatentMorph`. `LatentMorph` seamlessly integrates implicit reasoning into the autoregressive generation stream. (***Middle***) Dynamic Monitoring: a short-term condenser compresses recent hidden states $\mathbf{H}_{i-w:i}$ into a local memory, enabling the reason invoker $\mathcal{I}_{\text{invoker}}$ to adaptively trigger reasoning interventions. (***Bottom***) Latent Reasoning: upon invocation, a long-term condenser summarizes the global history $\mathbf{H}_{1:i}$ for $\text{UMM}_u$. The resulting latent thoughts $\mathbf{z}$ are transformed by the translator and shaper into control tokens $\mathbf{E}_{\text{ctrl}}$, which are injected directly into the generator's KV cache to steer subsequent synthesis without explicit text decoding.

$\mathbf{z} \in \mathbb{R}^d$ in continuous hidden states, which represent high-level insights and refinements derived from the reasoning.

To bridge the gap between reasoning and generation, a **latent translator** $\mathcal{T}_{\text{trans}}$ converts the latent thoughts $\mathbf{z}$ into generation-compatible control signals $\mathbf{c} \in \mathbb{R}^d$:

$$\mathbf{c} = \mathcal{T}_{\text{trans}}([\mathbf{z}; \mathbf{M}^{(l)}; \mathbf{p}]), \qquad (5)$$

where $\mathbf{p}$ denotes the prompt embedding of $T$ in $\text{UMM}_g$. Subsequently, a **latent shaper** $\mathcal{S}_{\text{shaper}}$ injects $\mathbf{c}$ into the generation pipeline by generating control tokens $\mathbf{E}_{\text{ctrl}} \in \mathbb{R}^{2B \times j \times d}$, which are inserted into the key-value (KV) cache of $\text{UMM}_g$, modifying subsequent token predictions:

$$\mathbf{KV}_{\text{new}} = \mathcal{S}_{\text{shaper}}(\mathbf{c}, \mathbf{KV}_{\text{old}}). \qquad (6)$$

This mechanism seamlessly integrates reasoning outputs into the generation process, ensuring implicit guidance without disrupting internal dynamics (*e.g.*, directly replace $\mathbf{p}$). We next detail the implementations of the condensers and invoker (Section §4.2) and the translator and shaper (Section §4.3), along with the training recipe (Section §4.4).

### 4.2. Learning to Invoke Reasoning with Visual Memory

Existing methods (Huang et al., 2025b; Qin et al., 2025a; Guo et al., 2025b) rely on fixed, predefined reasoning, where partially generated images are decoded and re-encoded into the understanding branch. In this section, `LatentMorph` aims to address two critical challenges: how to decide *when* reasoning should be invoked more dynamically; and how to efficiently represent the visual state of the generation process for reasoning. To this end, `LatentMorph` introduces the

reasoning invoker $\mathcal{I}_{\text{invoker}}$ as a dynamic and adaptive monitor, operating on a compact yet expressive representation of the generation state, constructed through short-term and long-term condensers $\mathcal{C}_{\text{short}}$ and $\mathcal{C}_{\text{long}}$.

**Short-Term Condenser for Invoker Monitoring.** The short-term condenser $\mathcal{C}_{\text{short}}$ plays a critical role in enabling the reasoning invoker to monitor the recent generation progress. At every window interval $w$, $\mathcal{C}_{\text{short}}$ compresses the hidden states of the most recent $w$ token hidden states $\mathbf{H}_{i-w:i}$ into a compact memory representation $\mathbf{M}^{(s)} \in \mathbb{R}^{n_s \times d}$, which encapsulates localized generation dynamics. To extract salient features from $\mathbf{H}_{i-w:i}$, $\mathcal{C}_{\text{short}}$ employs cross-attention with learnable latent queries $\mathbf{Q} \in \mathbb{R}^{n_s \times d}$:

$$\mathbf{M}^{(s)}, \mathbf{A} = \text{CA}(\mathbf{Q}, \mathbf{K} = \mathbf{H}_{i-w:i}, \mathbf{V} = \mathbf{H}_{i-w:i}), \quad (7)$$

where these memory tokens are further refined through a lightweight feedforward network, ensuring they retain meaningful information while remaining compact. To facilitate downstream decision-making, a pooled vector $\mathbf{m}^{(s)} = \text{Mean}(\mathbf{M}^{(s)})$ summarizes the short-term memory into a single representation. By focusing on the most recent generation window, $\mathcal{C}_{\text{short}}$ provides a concise and up-to-date view of the generation process, enabling the reasoning invoker to make informed and efficient decisions.

**Reasoning Invoker for Dynamic Invocation.** Unlike fixed-step reasoning schedules, $\mathcal{I}_{\text{invoker}}$, instantiated as a lightweight policy network (*e.g.*, MLP), adaptively determines *when* to invoke reasoning based on the evolving generation context, mimicking human-like reflection during creating. Specifically, $\mathcal{I}_{\text{invoker}}$ operates on a state feature

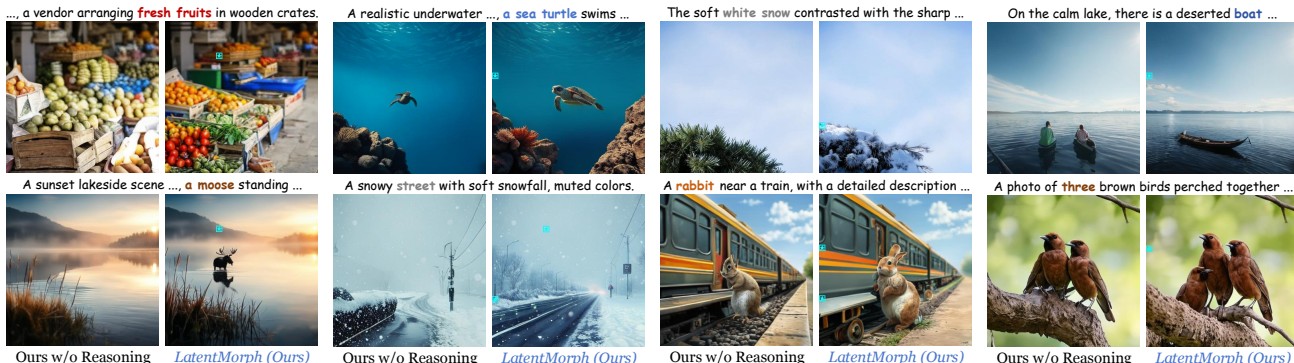

*Figure 3.* Case study of LatentMorph. The blue stars denote the adaptive reasoning invocations. These interventions align with critical semantic transitions, enabling LatentMorph to correct object omissions or counting errors observed in the baseline without reasoning.

vector $s_i$, which comprises multi-dimensional signals:

❖ **Semantic Consistency.** Semantic similarity $c_i = \cos(\mathbf{m}^{(s)}, \mathbf{p})$ between recent state and prompt embedding, which measures alignment with the user's intent.

❖ **Prediction Uncertainty.** Entropy of the token logits $u_i = (\mathbf{o}_i)$, capturing model confidence, where $\mathbf{o}_i$ represents the model's predicted probability distribution.

❖ **Temporal Dynamics.** Changes in semantic consistency $\Delta c_i = c_i - c_{i-w}$ and its variance $v_i = \text{Var}(c_{i-w:i})$, reflecting deviations and stability.

These signals are combined into $s_i = [c_i, u_i, \Delta c_i, v_i]$ and $\mathcal{I}_{\text{invoker}}$ computes the invocation decision as:

$$p_i = \pi_\theta(a_k = \text{REASON} \mid s_i) = \sigma(\mathcal{I}_{\text{invoker}}(s_i)), \quad (8)$$

where $a_k \in \{\text{CONTINUE}, \text{REASON}\}$ is sampled from a Bernoulli distribution parameterized by $p_i$. To encourage $\mathcal{I}_{\text{invoker}}$ to invoke reasoning only when necessary, we maximize an RL objective that balances task performance and reasoning efficiency, which avoids redundant reasoning steps, inspired by (Zhu et al., 2025a; Zhang et al., 2025a):

$$\max_\theta \mathbb{E}_{\tau \sim \pi_\theta} \left[ R(\tau) - \lambda \cdot \max(0, \bar{p}(\tau) - \bar{p}_{\text{ref}}) \right], \quad (9)$$

where $R(\tau)$ is the task reward (detailed in Section §4.4), $\bar{p}(\tau)$ is the average invoker probability over trajectory $\tau$, and $\bar{p}_{\text{ref}}$ is an adaptive reference level computed from high-reward trajectories in the batch.

**Long-Term Condenser for Reasoning Integration.** Once reasoning is triggered, the long-term condenser $\mathcal{C}_{\text{long}}$ summarizes the complete generation history $\mathbf{H}_{1:k}$ into a concise memory representation $\mathbf{M}^{(l)} \in \mathbb{R}^{n_l \times d}$. This long-term visual memory provides a high-level global overview of the generation trajectory, enabling $\text{UMM}_u$ to process visual evidence efficiently, rather than decode and re-encode. Different from $\mathcal{C}_{\text{short}}$, which captures localized trends, $\mathcal{C}_{\text{long}}$ employs streaming attention to handle arbitrarily long sequences by incrementally processing non-overlapping chunks of size $c$. For each chunk $\mathbf{H}_{t:t+c}$, cross-attention with learnable memory tokens $\mathbf{Q} \in \mathbb{R}^{n_l \times d}$ is applied as in Equation (7), where the memory updates incrementally:

$$\mathbf{M}^{(l)} = \text{Update}(\mathbf{M}^{(l)}, \mathbf{H}_{t:t+c}). \quad (10)$$

Here, only the top-$n_l$ most informative tokens are retained based on attention scores. Similar to $\mathcal{C}_{\text{short}}$, a pooled vector $\mathbf{m}^{(l)} = \text{Mean}(\mathbf{M}^{(l)})$ is computed to summarize the long-term memory. These representations are then passed to $\text{UMM}_u$, enabling reasoning directly in the hidden space.

### 4.3. Morphing Reasoning in Latent Space

A naive way to bridge the gap between $\text{UMM}_u$ and $\text{UMM}_g$ might directly map the latent thoughts $\mathbf{z}$ into the prompt embedding $\mathbf{p}$ space, akin to explicit reasoning paradigms that replace prompts (Guo et al., 2025b). However, such a direct replacement treats reasoning as static, neglecting the dynamic nature of autoregressive generation and undermining $\mathbf{p}$'s role as a guiding signal that interacts with evolving generation states. In this section, LatentMorph introduces the latent translator $\mathcal{T}_{\text{trans}}$ and latent shaper $\mathcal{S}_{\text{shaper}}$, which transform $\mathbf{z}$ into actionable guidance and seamlessly inject it into the generation pipeline, preserving autoregressive consistency and enabling adaptive refinement.

**Latent Translator for Guidance Generation.** The latent translator $\mathcal{T}_{\text{trans}}$ is responsible for converting $\mathbf{z}$, along with long-term visual memory $\mathbf{m}^{(l)}$ and the prompt embedding $\mathbf{p}$, into generation-compatible control signals $\mathbf{c}$, as formalized in Equation (5). Internally, $\mathcal{T}_{\text{trans}}$ employs a lightweight MLP with residual connections and a gating mechanism to adaptively filter noise and retain salient information:

$$\mathbf{c}' = \text{MLP}([\mathbf{z}; \mathbf{m}^{(l)}; \mathbf{p}]), \quad \mathbf{g} = \tanh(\text{Linear}(\mathbf{c}')), \quad (11)$$

where $\mathbf{c} = \mathbf{c}' \odot \mathbf{g}$. This design allows $\mathcal{T}_{\text{trans}}$ to adaptively modulate the contribution of reasoning outputs, ensuring that only salient features influence the generation process. By incorporating $\mathbf{m}^{(l)}$, $\mathcal{T}_{\text{trans}}$ leverages historical generation context, enabling reasoning to influence the process in a globally consistent manner.

**Latent Shaper for Control Injection.** Inspired by the use of control tokens in conditional generation tasks, $\mathcal{S}_{\text{shaper}}$ extends the concept to dynamically inject reasoning-derived guidance during the generation process. Specifically, $\mathcal{S}_{\text{shaper}}$ dynamically transforms $\mathbf{c}$ into a sequence of control tokens $\mathbf{E}_{\text{ctrl}} \in \mathbb{R}^{2B \times j \times d}$, where $B$ is the batch size, $j$ is the number of control tokens, and $d$ is the hidden dimension. $\mathbf{E}_{\text{ctrl}}$ is

*Table 1.* Evaluation on GenEval, and T2I-CompBench. The best and second best results are highlighted.

| Method | GenEval | | | | T2I-CompBench | | | | | | |
|---|---|---|---|---|---|---|---|---|---|---|---|
| | Two Object↑ | Position↑ | Color Attribute↑ | Overall↑ | Color↑ | Shape↑ | Texture↑ | Spatial↑ | Non-Spatial↑ | Complex↑ | Overall↑ |
| Vanilla | 0.89 | 0.79 | 0.66 | 0.80 | 63.59 | 35.28 | 49.36 | 20.61 | 30.85 | 35.59 | 39.21 |
| SFT | 0.87 | 0.79 | 0.66 | 0.79 | 64.23 | 34.56 | 49.46 | 20.98 | 31.55 | 35.80 | 39.43 |
| GRPO | 0.90 | 0.80 | 0.66 | 0.82 | 67.90 | 36.31 | 52.13 | 23.01 | 31.29 | 39.04 | 41.61 |
| Self-CoT (Deng et al., 2025) | 0.90 | 0.78 | 0.70 | 0.83 | 68.19 | 37.89 | 54.10 | 21.90 | 30.00 | 44.01 | 42.68 |
| T2I-R1 (Jiang et al., 2025) | 0.91 | 0.76 | 0.65 | 0.79 | 81.30 | 58.52 | 72.41 | 33.78 | 30.90 | 39.93 | 52.81 |
| TIR (Khan et al., 2025) | 0.93 | 0.84 | 0.68 | 0.84 | 68.92 | 49.12 | 60.10 | 21.77 | 31.21 | 40.12 | 45.21 |
| T2I-Copilot (Chen et al., 2025a) | 0.94 | 0.86 | 0.66 | 0.85 | 67.42 | 47.82 | 61.34 | 22.12 | 30.41 | 41.85 | 45.16 |
| MILR (Mi et al., 2025) | 0.96 | 0.98 | 0.91 | 0.95 | 85.08 | 51.17 | 69.49 | 46.13 | 30.78 | 36.84 | 53.25 |
| TwiG-ZS (Guo et al., 2025b) | 0.94 | 0.84 | 0.67 | 0.86 | 73.11 | 41.55 | 64.77 | 21.98 | 30.90 | 48.16 | 46.75 |
| TwiG-RL (Guo et al., 2025b) | - | - | - | - | 82.49 | 61.28 | 73.19 | 34.06 | 31.99 | 54.45 | 56.24 |
| LatentMorph (Ours) | 0.97 | 0.98 | 0.92 | 0.96 | 84.04 | 69.46 | 79.89 | 50.93 | 39.27 | 63.60 | 64.53 |

then inserted into the KV cache of $\mathrm{UMM}_g$, modifying subsequent token predictions without altering the autoregressive structure or occupying prediction positions:

$$\mathbf{E}_{\mathrm{ctrl}} = \mathcal{S}_{\mathrm{shaper}}(\mathbf{c}), \mathbf{KV}_{\mathrm{new}} = \mathrm{Update}(\mathbf{KV}_{\mathrm{old}}, \mathbf{E}_{\mathrm{ctrl}}). \quad (12)$$

By injecting control tokens directly into the KV cache, $\mathcal{S}_{\mathrm{shaper}}$ enables reasoning-derived guidance to influence subsequent token predictions implicitly, without requiring explicit decoding or disrupting the autoregressive dynamics. This approach ensures that the generation process remains efficient and cognitively aligned, dynamically adapting to reasoning outputs as needed.

### 4.4. Two-Stage Training of LatentMorph

**Supervised Fine-Tuning (SFT).** We train the long-term modules, *i.e.*, the condenser $\mathcal{C}_{\mathrm{long}}$, translator $\mathcal{T}_{\mathrm{trans}}$, and shaper $\mathcal{S}_{\mathrm{shaper}}$, using 20k text-image pairs from midjourney-prompts (vivym, 2023). Each image is associated with a single randomly triggered reasoning step during generation. Unlike prior works on interleaving reasoning, *e.g.*, (Guo et al., 2025b; Gu et al., 2025), or latent reasoning, *e.g.*, (Li et al., 2025a; Dong et al., 2025), which require extensive process-level supervision or curated datasets for each reasoning step, our approach relies solely on a standard cross-entropy (CE) loss for autoregressive image generation, achieving adaptive learning without additional supervision.

**Reinforcement Learning (RL).** The reasoning invoker $\mathcal{I}_{\mathrm{invoker}}$ and short-term condenser $\mathcal{C}_{\mathrm{short}}$ are optimized using GRPO (Guo et al., 2025a) with a group size of 8, incorporating the penalty mechanism introduced in Equation (9) for adaptive reasoning invocation. Following TwiG (Guo et al., 2025b), we use the training set prompts from T2I-CompBench (Huang et al., 2023) for training. For reward models, we adopt HPS-v2.1 (Wu et al., 2023) and CLIP score (Radford et al., 2021), as in DanceGRPO (Xue et al., 2025). All experiments are conducted on 8 NVIDIA H200 GPUs, with additional details provided in Appendix §A.

## 5. Experiments

In this section, we conduct extensive experiments to answer the following research questions:

**RQ1:** Can LatentMorph outperform explicit interleaving reasoning paradigms?

**RQ2:** Is latent reasoning by LatentMorph effective in understanding abstract and complex user prompts?

**RQ3:** Does LatentMorph's adaptive latent reasoning invocation facilitate more efficient generation?

**RQ4:** Does LatentMorph align better with human cognitive processes in artistic creation?

### 5.1. Experimental Settings

**Baselines.** We apply LatentMorph to the advanced pure autoregressive UMM Janus-Pro (Chen et al., 2025c). We focus our comparisons on the following ten baseline strategies: **(I) Generation-only methods**: the vanilla model, SFT, GRPO (Guo et al., 2025a); **(II) Reason-before/after-generation**: Self-CoT (Deng et al., 2025), T2I-R1 (Jiang et al., 2025), TIR (Khan et al., 2025), T2I-Copilot (Chen et al., 2025a), MILR (Mi et al., 2025); and **(III) Reason-while-generation**: TwiG-ZS, TwiG-RL (Guo et al., 2025b). Details on baselines are provided in Appendix §B.1.

**Evaluations.** We conduct evaluations across five benchmarks focusing on three key aspects: (*i*) *General*: GenEval (Ghosh et al., 2023); (*ii*) *Compositional*: T2I-CompBench (Huang et al., 2023), T2I-CompBench++ (Huang et al., 2025a); and (*iii*) *Complex*: WISE (Niu et al., 2025), IPV-Txt (Bai et al., 2025). Details are placed in Appendix §B.2.

### 5.2. Main Results

To answer **RQ1**, we conduct comprehensive comparisons against ten baselines on general and compositional benchmarks in Tables 1, 3, along with qualitative results shown in Figures 4, 8 and 9. Key observations are summarized:

**Obs.❶ LatentMorph establishes superior fidelity in both general and compositional generation.** As shown in Table 1, LatentMorph consistently outperforms state-of-the-art reasoning-augmented paradigms overall. Most notably in the challenging Non-Spatial category of T2I-CompBench, LatentMorph surpasses the leading reason-while-generation baseline TwiG-RL by a significant mar-

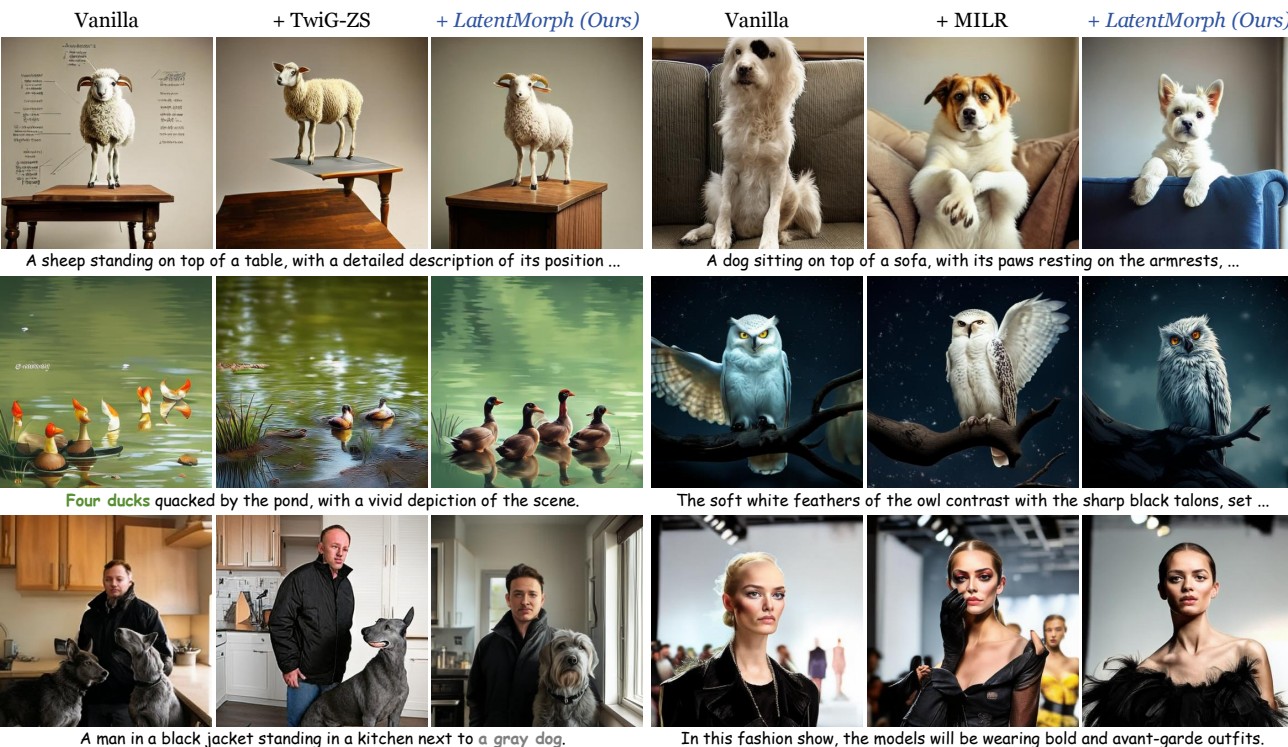

| Vanilla | + TwiG-ZS | + *LatentMorph (Ours)* | Vanilla | + MILR | + *LatentMorph (Ours)* |

*Figure 4.* Qualitative comparison of LatentMorph.

gin of 7.28%. This advantage widens further against *iterative* reason-before/after strategies; LatentMorph exceeds T2I-Copilot and MILR by 8.86% and 8.49% respectively, achieving better alignment without their heavy computational overhead. We attribute this performance leap to the continuous nature of our latent reasoning: unlike TwiG or T2I-Copilot, which compress intermediate thoughts into discrete text, LatentMorph retains rich semantic cues within continuous high-dimensional states. This robustness extends to fine-grained tasks in T2I-CompBench++ in Table 3, where LatentMorph outperforms TwiG-RL by 8.13% in 3D-Spatial and 7.32% overall. Qualitative results in Figures 4, 8 and 9 visually reinforce these findings, demonstrating LatentMorph's ability to resolve complex attributes where baselines frequently suffer from hallucination.

## 5.3. Effectiveness Analysis

To answer **RQ2**, we extend our evaluation to the more abstract and cognitively demanding benchmark, *i.e.*, WISE, as well as the idea of "impossible prompting" in IPV-Txt, as shown in Figure 5 (*Left & Middle*). To further isolate the efficacy of latent *vs.* explicit reasoning, we introduce a baseline "LatentMorph w/o latent", where the latent thought $z$ is forced through a decode-re-encode bottleneck before control injection, and the attention differences are visualized in Figure 5 (*Right*). Our observations are:

**Obs.❷ Latent reasoning captures ineffable semantics lost in textual decoding.** Quantitative results in Figure 5 (*Left*) reveal a consistent performance hierarchy. Notably,

on the IPV-Txt benchmark, which involves counterintuitive physical dynamics, explicit reasoning methods, *e.g.*, TwiG-ZS, struggle, likely because natural language is insufficient to precisely describe complex concepts. Even our own variant, w/o latent, suffers a performance drop solely due to the information loss from decoding thoughts into text. This is visually explained in Figure 5 (*Right*). The differential heatmaps show that LatentMorph activates attention in regions characterized by subtle textures and lighting, attributed to the semantically dense but difficult to verbalize explicitly. This confirms that performing reasoning entirely in the continuous latent space preserves critical, non-verbalizable visual cues essential for generation.

## 5.4. Efficiency Analysis

To answer **RQ3**, we further quantify the inference latency and total token consumption of LatentMorph against representative baselines on T2I-CompBench, as demonstrated in Figure 6 (*Left*). Our findings are summarized as follows:

**Obs.❸ LatentMorph is a time- and token-efficient autoregressive image generation enhancer.** As illustrated in Figure 6 (*Left*), LatentMorph incurs negligible computational overhead compared to the vanilla baseline, while demonstrating significant efficiency gains over explicit reasoning paradigms. Iterative methods like MILR require multiple full-generation cycles to search for optimal latents, and reason-while-generating frameworks like TwiG-ZS are bottlenecked by the frequent decoding of partial images and the verbose textual thoughts. In contrast, LatentMorph

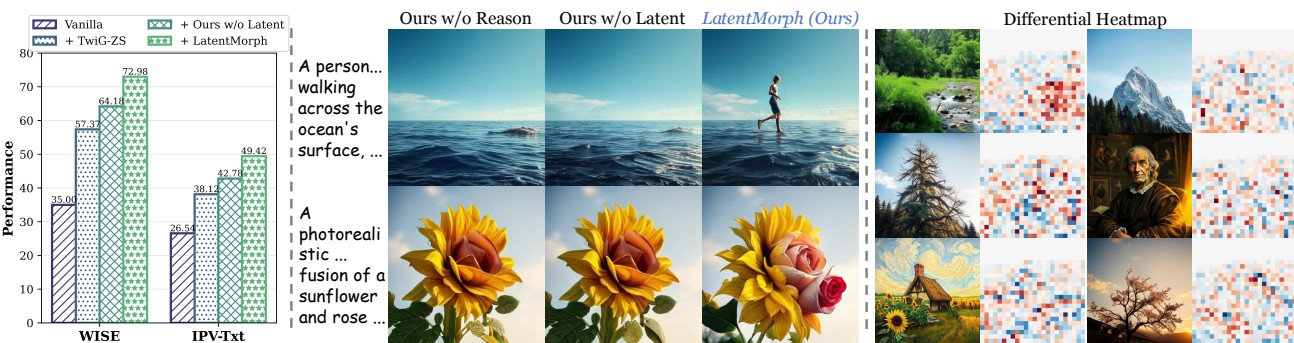

Figure 5. (*Left*) Evaluation results on WISE and IPV-Txt. (*Middle*) Qualitative examples on "impossible prompts" of IPV-Txt. (*Right*) Differential heatmap between latent and explicit reasoning, highlighting the information loss incurred by discrete text thoughts.

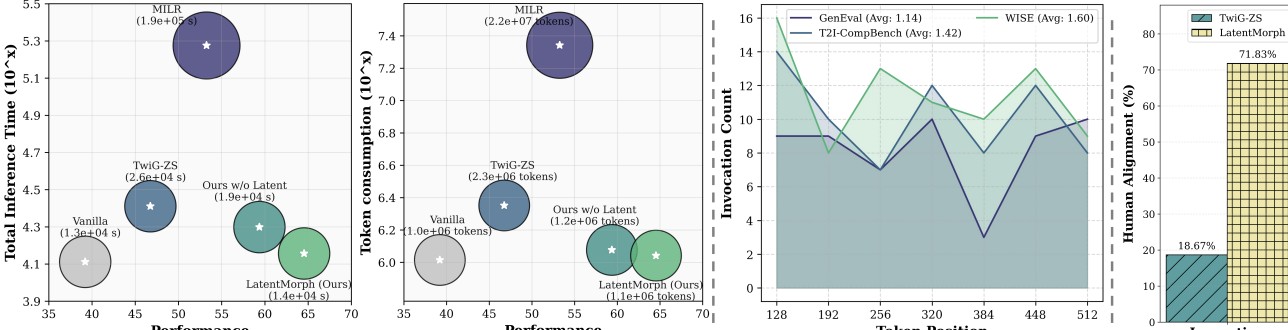

Figure 6. (*Left*) Inference time and token consumption of `LatentMorph` *vs.* baselines on T2I-CompBench. (*Middle*) Reasoning invocation across GenEval, T2I-CompBench, and WISE. (*Right*) User study on the timing of reasoning intervention.

operates as a seamless single-pass stream. By condensing history into compact visual memory and injecting control signals directly, we eliminate the expensive steps of pixel-space decoding and explicit text tokenization. Notably, even our `LatentMorph` w/o latent variant outperforms fixed-step baselines (*e.g.*, TwiG-ZS) in efficiency. This further validates the design of our adaptive invoker, which intelligently triggers reasoning only when necessary, avoiding redundant computation on well-aligned generation steps.

### 5.5. Framework Analysis

To answer **RQ4**, we further analyze the frequency and positioning of reasoning invocations across benchmarks (each with 50 prompts) in Figure 6 (*Middle*). Complementing this, we conduct a user study to evaluate the alignment between `LatentMorph`'s invocation decisions and human cognitive intuition, summarized in Figure 6 (*Right*), alongside a case study in Figure 3. Our findings are summarized below:

**Obs.❹ `LatentMorph` mimics the adaptive rhythm of human creative cognition.** Unlike fixed-step invocation paradigms, *e.g.*, TwiG, that force reasoning at rigid intervals regardless of context, `LatentMorph` exhibits a context-aware invocation pattern. As shown in Figure 6 (*Middle*), the invocation frequency correlates positively with task complexity, *i.e.*, triggering sparsely for simple prompts (avg. 1.14 on GenEval) and more frequently for abstract reasoning tasks (avg. 1.60 on WISE). This aligns with human creative processes: pausing to reflect only when encounter-

ing bottlenecks. The user study in Figure 6 (*Right*) further corroborates this, where human evaluators rated our adaptive timing as significantly more natural and necessary compared to fixed baselines. Case studies in Figure 3. visually demonstrate this behavior, showing the model intervening precisely at critical compositional transitions.

**Ablation Study.** To further confirm the superiority of our learned invoker policy, Table 4 in Appendix §C shows that our adaptive strategy consistently outperforms both random and fixed-step injections, confirming that *when* to reason is as critical as *how*. Comprehensive sensitivity analyses covering the condensers, translator, shaper, as well as invoker hyperparameters are detailed in Appendix §C.

## 6. Conclusion

In this work, we present `LatentMorph`, a framework that integrates implicit latent reasoning into autoregressive text-to-image generation. By bypassing the bottlenecks of explicit textual decoding, `LatentMorph` enables continuous reasoning and refinement within high-dimensional latent spaces. Leveraging visual memory condensation, latent translation, and adaptive control injection, it achieves a strong balance between generation fidelity and efficiency. Additionally, the RL-based invoker dynamically aligns the generation process with human-like cognitive rhythms, intervening only when necessary. This paradigm shift from explicit reasoning to implicit intuition sets the foundation for the next generation of cognitively aligned and creatively robust visual generation.

## Impact Statement

This paper presents LatentMorph, a framework aimed at enhancing the fidelity and efficiency of text-to-image generation. On the positive side, LatentMorph significantly reduces the computational overhead associated with reasoning-augmented generation, contributing to the development of more energy-efficient and environmentally sustainable AI systems. Additionally, by better aligning with human cognitive processes, it serves as a more intuitive tool for artistic expression and creative workflows.

However, as with all advancements in high-fidelity generative models, there is a potential for misuse in creating photorealistic, misleading content or deepfakes. It is important to note that LatentMorph operates by optimizing the latent states of Janus-Pro as post-training and does not introduce new pre-training data. Consequently, LatentMorph inherits both the safety guardrails and the potential biases present in the underlying base model. We strongly encourage the deployment of such technologies in conjunction with robust safety filters and watermarking mechanisms.

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

# A. Training Details of `LatentMorph`

## A.1. More Details of Two-Stage Training

We employ a two-stage training pipeline to progressively equip the model with latent reasoning capabilities. We provide more details of the two training stages in this section:

### A.1.1. STAGE I: SUPERVISED FINE-TUNING (SFT) FOR LATENT CONTROL

The primary goal of the SFT stage is to train the long-term condenser ($\mathcal{C}_{\text{long}}$), latent translator ($\mathcal{T}_{\text{trans}}$), and latent shaper ($\mathcal{S}_{\text{shaper}}$) to generate effective control signals from visual history.

**Random Injection Strategy.** Since we do not have ground-truth data indicating when to reason, we employ a randomized injection strategy during SFT. For a given text-image pair $(T, I)$, we randomly sample an intervention step $k$ from a uniform distribution $k \sim \mathcal{U}[k_{min}, k_{max}]$, where we set $[k_{min}, k_{max}] = [150, 450]$ to avoid edge effects at the very beginning or end of generation.

**Forward Process & Loss.** At step $k$, we extract the history $\mathbf{H}_{1:k}$ and compute the control tokens $\mathbf{E}_{\text{ctrl}}$ via the long-term reasoning branch:

$$\mathbf{m}^{(l)} = \mathcal{C}_{\text{long}}(\mathbf{H}_{1:k}), \quad \mathbf{z} = \text{UMM}_u(\mathbf{m}^{(l)}, T), \quad \mathbf{E}_{\text{ctrl}} = \mathcal{S}_{\text{shaper}}(\mathcal{T}_{\text{trans}}([\mathbf{z}; \mathbf{m}^{(l)}; \mathbf{p}])). \tag{13}$$

Crucially, to maintain training efficiency and autoregressive consistency, we inject $\mathbf{E}_{\text{ctrl}}$ directly into the KV cache of $\text{UMM}_g$ at step $k$. This allows us to compute the loss for all subsequent tokens $x_{k+1:|X|}$ in a single forward pass without physically splicing the sequence. The optimization objective is the standard negative log-likelihood over the image tokens, conditioned on the latent injection:

$$\mathcal{L}_{SFT} = -\sum_{t=1}^{|X|} \log p_\theta(x_t | x_{<t}, T, \mathbb{I}_{t>k} \cdot \mathbf{E}_{\text{ctrl}}), \tag{14}$$

where $\mathbb{I}_{t>k}$ indicates that the control tokens only influence predictions after step $k$.

### A.1.2. STAGE II: REINFORCEMENT LEARNING (RL) FOR ADAPTIVE INVOCATION

In the second stage, we freeze the modules trained in SFT and focus on optimizing the reasoning invoker ($\mathcal{I}_{\text{invoker}}$) and short-term condenser ($\mathcal{C}_{\text{short}}$). We utilize Group Relative Policy Optimization (GRPO) (Guo et al., 2025a) to encourage the model to invoke reasoning only when necessary to improve alignment.

**State Space Construction.** To enable the policy $\pi_\theta$ to make informed decisions, we construct a comprehensive state representation $s_i$ at each check interval (every $w = 32$ tokens). The state $s_i$ concatenates four specific signals extracted from the generation stream:

- **Semantic Consistency**: Cosine similarity between the short-term memory and prompt: $c_i = \cos(\mathbf{m}^{(s)}, \mathbf{p})$.

- **Temporal Dynamics**: The change in consistency $\Delta c_i = c_i - c_{i-w}$.

- **Stability**: The variance of consistency scores over the recent window $v_i = \text{Var}(c_{i-w:i})$.

- **Uncertainty**: The entropy of the current token distribution $u_i = \mathcal{H}(p(x_i))$.

**GRPO Formulation.** For each prompt $T$, we generate a group of $G = 8$ trajectories $\{\tau_1, ..., \tau_G\}$. For each trajectory, $\mathcal{I}_{\text{invoker}}$ samples an action $a_t \in \{\text{CONTINUE}, \text{REASON}\}$ at each interval. The policy gradient is estimated as:

$$\nabla_\theta J = \frac{1}{G} \sum_{j=1}^{G} \sum_t \frac{R(\tau_j) - \bar{R}}{\sigma_R} \nabla_\theta \log \pi_\theta(a_{t,j} | s_{t,j}), \tag{15}$$

where $\bar{R}$ and $\sigma_R$ are the mean and standard deviation of rewards within the group.

**Reward Shaping and Penalty.** The trajectory reward $R(\tau)$ is a weighted sum of the CLIP score ($R_{CLIP}$) (Radford et al., 2021) and Human Preference Score (HPS-v2.1) ($R_{HPS}$) (Wu et al., 2023), with weights $w_{clip} = 1.0$ and $w_{hps} = 1.0$ respectively. To prevent the model from trivially invoking reasoning at every step, we enforce the penalty term (Equation 9) with an adaptive threshold. Specifically, we penalize the policy if the average invocation probability $\bar{p}$ deviates from a reference level derived from high-quality samples in the group.

## A.2. More Details of Parameter Configurations

*Table 2.* Hyperparameter configurations for `LatentMorph`.

| Setting | Hyperparameter | Value |
|---|---|---|
| *Model Architecture* | | |
| $\mathcal{C}_{\text{short}}$ | memory tokens ($n_s$) | 4 |
| | attention heads | 4 |
| | MLP ratio | 2.0 |
| $\mathcal{C}_{\text{long}}$ | memory tokens ($n_l$) | 8 |
| | attention heads | 8 |
| | chunk size ($c$) | 64 |
| | streaming accumulation | FP32 |
| $\mathcal{T}_{\text{trans}}$ | hidden ratio | 2.0 |
| | max scale (gate) | 1.0 |
| $\mathcal{S}_{\text{shaper}}$ | control tokens ($j$) | 4 |
| $\mathcal{I}_{\text{invoker}}$ | input dimension | 4 |
| | hidden dimension | 32 |
| | layers | 2 |
| *Stage I: Supervised Fine-Tuning (SFT)* | | |
| **Optimization** | optimizer | AdamW ($\beta_1 = 0.9, \beta_2 = 0.999$) |
| | learning rate | $1e-4$ |
| | weight decay | 0.0 |
| | global batch size | 64 |
| **Injection** | random injection range | $[150, 450]$ |
| **Data** | image resolution | $384 \times 384$ |
| *Stage II: Reinforcement Learning (RL)* | | |
| **Optimization** | algorithm | GRPO |
| | group size ($G$) | 8 |
| | learning rate | $1e-5$ |
| | advantage clipping | $\pm 5.0$ |
| **Policy** | check interval ($w$) | 64 |
| | entropy coefficient | 0.001 |
| | penalty lambda ($\lambda$) | 0.2 |
| **Reward** | weights (CLIP / HPS) | 1.0/1.0 |

## A.3. Compatibility Analysis

In this section, we provide a further analysis to demonstrate that `LatentMorph` is theoretically compatible with various pure autoregressive generator paradigms. Our framework relies on the universal abstraction of Transformer-based autoregressive generation, making it model-agnostic.

### A.3.1. UNIVERSAL AUTOREGRESSIVE FORMULATION

Consider a generic autoregressive image generator $\text{UMM}_g$ parameterized by $\theta$. Given a condition $T$, the generation of an image sequence $X = (x_1, \ldots, x_{|X|})$ is modeled as a joint probability decomposed into conditional probabilities:

$$p_\theta(X|T) = \prod_{i=1}^{|X|} p_\theta(x_i|x_{<i}, T) \tag{16}$$

In Transformer-based architectures, the probability of the $i$-th token is computed based on the hidden state $\mathbf{h}_i^L$ of the final layer $L$:

$$\mathbf{h}_i^l = \text{Attention}(\mathbf{h}_i^{l-1}, \mathbf{H}_{<i}^{l-1} \cup \mathbf{H}_T), \quad p(x_i|\cdot) = \text{Softmax}(W_{\text{vocab}}\mathbf{h}_i^L), \tag{17}$$

where $\mathbf{H}_{<i}$ represents the history of visual hidden states (cached as KV pairs) and $\mathbf{H}_T$ represents the condition embeddings.

A.3.2. INTERFACE COMPATIBILITY

LatentMorph interacts with $\text{UMM}_g$ solely through the latent space interface, defined by the read operation ($\mathcal{C}_{\text{short}}$, $\mathcal{C}_{\text{long}}$) and the write operation ($\mathcal{S}_{\text{shaper}}$). We define the compatibility conditions as follows:

**State Readout Compatibility (Condenser).** The condenser modules ($\mathcal{C}_{\text{short}}$ and $\mathcal{C}_{\text{long}}$) operate on the sequence of hidden states $\mathbf{H}_{1:i}$. For any AR generator with hidden dimension $d_g$, we can introduce a linear projection $W_{\text{proj}} \in \mathbb{R}^{d_g \times d}$ to map the generator's state space to the reasoning space:

$$\mathbf{M}^{(s)} = \text{CrossAttn}(Q, K = W_{\text{proj}}\mathbf{H}_{i-w:i}, V = W_{\text{proj}}\mathbf{H}_{i-w:i}). \tag{18}$$

Since $\mathcal{C}$ relies on Cross-Attention, it is invariant to the specific architecture (*e.g.*, number of layers or heads) of $\text{UMM}_g$, provided that $\mathbf{H}$ is accessible.

**Control Injection Compatibility (Shaper).** The shaper $\mathcal{S}_{\text{shaper}}$ influences the generation by injecting control signals $\mathbf{E}_{\text{ctrl}} \in \mathbb{R}^{j \times d}$ into the attention mechanism. Mathematically, this modifies the attention context for subsequent steps $t > k$:

$$\text{Attention}(q_t, K', V') = \text{Softmax}\left(\frac{q_t[K_{<t}; K_{\text{ctrl}}]^\top}{\sqrt{d}}\right)[V_{<t}; V_{\text{ctrl}}], \tag{19}$$

where $K_{\text{ctrl}}, V_{\text{ctrl}}$ are the keys and values derived from $\mathbf{E}_{\text{ctrl}}$. Crucially, this injection maintains absolute positional consistency, making it strictly compatible with Rotary Positional Embeddings (RoPE). The control signals act as a virtual context attached to the intervention step $k$ without shifting the positional indices of subsequent tokens ($x_{t>k}$), thereby preserving the internal relative distance dynamics. This formulation shows that `LatentMorph` acts as a seamless extension of the conditioning set without altering model weights $\theta$:

$$p_\theta(x_t|x_{<t}, T, \mathbf{E}_{\text{ctrl}}) \approx p_\theta(x_t|x_{<t}, \{T, \texttt{LatentThoughts}\}) \tag{20}$$

Thus, generators that utilize a Key-Value cache mechanism is compatible with our injection method without structural changes.

A.3.3. ADAPTATION TO EXTERNAL-LOOP PARADIGMS

Based on the formulation above, we analyze the external-loop paradigm (*i.e.*, decoupled autoregressive generators) mentioned in Section §1. In this setting, the generator $\text{UMM}_g$ (*e.g.*, LlamaGen (Sun et al., 2024)) is separate from the reasoning model $\text{UMM}_u$ (*e.g.*, a VLM). The generator operates on pixel/token space $X$, while the reasoner operates on a separate semantic space. `LatentMorph` bridges this gap via the learnable adapters trained in SFT: ❶ $\mathcal{C}_{\text{long}}$ acts as a visual encoder, projecting the generator's specific hidden states $\mathbf{H}_g$ into the reasoner's input space $d_u$. ❷ $\mathcal{S}_{\text{shaper}}$ acts as a control adapter, projecting the reasoner's output $\mathbf{c}$ back into the generator's dimension $d_g$ and format. This demonstrates that `LatentMorph` essentially functions as a differentiable neural interface, enabling "System-2" reasoning on any "System-1" autoregressive generator, regardless of whether they share a backbone.

# B. Experimental Details of `LatentMorph`

## B.1. More Details of Baseline Implementations

In this section, we provide detailed implementation configurations for each baseline method included in our comparison:

- **SFT & GRPO**: To ensure a fair comparison, we directly fine-tune the vanilla model (*i.e.*, Janus-Pro (Chen et al., 2025c)) using the same data and training configuration as `LatentMorph`, such as the dual reward setting employed in GRPO.

- **Self-CoT** (Deng et al., 2025): We adopt the Self-CoT strategy from Bagel (Deng et al., 2025) as a representative reason-before-generation paradigm. Since Bagel utilizes a diffusion head incompatible with our autoregressive setting, we implement this strategy directly on our vanilla model to facilitate a direct comparison.

- **T2I-R1** (Jiang et al., 2025): T2I-R1 represents a sophisticated reason-before-generation paradigm specifically designed for Janus-Pro, featuring a CoT RL framework with two-level CoT data construction. We directly employ their released pre-trained model for evaluation on benchmarks such as GenEval.

- **TIR** (Khan et al., 2025): TIR operates as a test-time optimization strategy following the reason-after-generation paradigm. As the original implementation did not target Janus-Pro, we adapted and deployed TIR on our vanilla model for our experiments.

- **T2I-Copilot** (Chen et al., 2025a): T2I-Copilot is a training-free multi-agent system that optimizes T2I generation by

interleaving between reason-before and reason-after paradigms. Similar to TIR, we adapted this framework to function with Janus-Pro to enable comparative evaluation.

- **MILR** (Mi et al., 2025): MILR represents a distinct test-time latent optimization strategy within the internal-loop paradigm. Following LatentSeek (Li et al., 2025b), it approaches reasoning as a latent search process rather than a generative one. Specifically, MILR iteratively optimizes intermediate latent representations during inference by maximizing feedback from an external reward model via policy gradients. Unlike `LatentMorph`, which learns when to reason in a single forward pass, MILR relies on computationally intensive iterative search to locate optimal latent states. We adopt this strategy regarding it as a paradigm of interleaved reason-before and reason-after.

- **TwiG-ZS & TwiG-RL** (Guo et al., 2025b): TwiG is a recent reason-while-generating framework built upon Janus-Pro. Our comparison includes both the zero-shot (ZS) and GRPO-trained versions. Due to the unavailability of the official code, we reproduced TwiG-ZS following the implementation details in the paper. For TwiG-RL, as the training data is also proprietary, we directly report the results on T2I-CompBench(++) as presented in their original publication.

## B.2. More Details of Evaluation Benchmarks

In this section, we provide further details on the benchmarks used in our evaluations:

- **GenEval** (Ghosh et al., 2023): GenEval is a widely adopted benchmark designed to evaluate the general alignment capabilities of T2I models across a diverse range of prompts.

- **T2I-CompBench** (Huang et al., 2023) **& T2I-CompBench++** (Huang et al., 2025a): T2I-CompBench focuses specifically on evaluating compositional generation capabilities. We also include its enhanced version, T2I-CompBench++, which expands the evaluation scope to include additional dimensions such as numeracy.

- **WISE** (Niu et al., 2025): WISE is a recently proposed benchmark that emphasizes world knowledge. It features prompts that are significantly more abstract and complex than those in standard datasets, making it particularly suitable for evaluating the capabilities of reasoning-augmented T2I generation models.

- **IPV-Txt** (Bai et al., 2025): IPV-Txt focuses on scenarios involving counter-intuitive physical phenomena, aiming to test whether models grasp underlying physical laws rather than merely fitting training data distributions. Although originally designed for text-to-video generation, we select a subset of prompts applicable to the image domain to evaluate the model's understanding of abstract concepts and physical constraints. Following (Chen et al., 2025b), we adopt the Impossible Prompt Following (IPF) from (Bai et al., 2025) as the evaluation metric, which measures the alignment between generated images and the semantic intent of impossible prompts, and employ GPT-4o to perform judgments.

## B.3. System Prompt for Reasoning Core

In this section, we detail the system prompt for the reasoning branch $\text{UMM}_u$ whenever the invoker triggers the reasoning process. Given that the primary focus of `LatentMorph` lies in the mechanism of interleaving implicit latent reasoning rather than exploring complex prompt engineering strategies, we make it *as simple as possible to be practical* here.

---

**Reasoning Prompt for $\text{UMM}_u$**

```
"""You are an image generation optimizer. Your task is to monitor the status of generated images and
provide optimization guidance.

You are given the currently generated image representation:

**{rep_ids}**

This representation corresponds to the original generation objective:

**{base_prompt}**

Please carefully analyze the current state of images generated and deliberate on how subsequent
generation should be optimized to correct potential biases while maintaining semantic, structural, and
visual consistency.

Do not output any intermediate reasoning, verification, or explanations. Implicitly validate consistency
and generate a prompt that enables continued image generation. Output only the continuation prompt used to
generate the remaining image tokens.

"""
```

---

*Table 3.* Evaluation on T2I-CompBench++ (Huang et al., 2025a). We highlight the best and second best results.

| Method | Color↑ | Shape↑ | Texture↑ | 2D-Spatial↑ | 3D-Spatial↑ | Non-Spatial↑ | Numeracy↑ | Comeplex↑ |
|---|---|---|---|---|---|---|---|---|
| Vanilla | 63.59 | 35.28 | 49.36 | 20.61 | 32.94 | 30.85 | 41.52 | 35.59 |
| SFT | 64.23 | 34.56 | 49.46 | 20.98 | 32.90 | 31.55 | 41.09 | 35.80 |
| GRPO | 67.90 | 36.31 | 52.13 | 23.01 | 31.78 | 31.29 | 42.01 | 39.04 |
| Self-CoT (Deng et al., 2025) | 68.19 | 37.89 | 54.10 | 21.90 | 34.09 | 30.00 | 39.89 | 44.01 |
| T2I-R1 (Jiang et al., 2025) | 81.30 | 58.52 | 72.41 | 33.78 | 34.23 | 30.90 | 45.32 | 39.93 |
| TIR (Khan et al., 2025) | 68.92 | 49.12 | 60.10 | 21.77 | 35.48 | 31.21 | 41.20 | 40.12 |
| T2I-Copilot (Chen et al., 2025a) | 67.42 | 47.82 | 61.34 | 22.12 | 34.02 | 30.41 | 43.44 | 41.85 |
| MILR (Mi et al., 2025) | 85.08 | 51.17 | 69.49 | 46.13 | 38.08 | 30.78 | 60.21 | 36.84 |
| TwiG-ZS (Guo et al., 2025b) | 73.11 | 41.55 | 64.77 | 21.98 | 33.68 | 30.90 | 36.58 | 48.16 |
| TwiG-RL (Guo et al., 2025b) | 82.49 | 61.28 | 73.19 | 34.06 | 38.87 | 31.99 | 61.93 | 54.45 |
| LatentMorph (**Ours**) | 84.04 | 69.46 | 79.89 | 50.93 | 47.00 | 39.56 | 62.33 | 63.60 |

## B.4. Captions of Figure 5

In this section, we provide the detailed prompts in Figure 5:

- ***Middle***: (**1st**) *"A person defies physics by walking confidently across the ocean's surface, their feet remaining completely dry as if treading on invisible solid ground. The calm seawater, which should naturally engulf anyone stepping on it, appears to have transformed into a firm platform beneath their feet. The surrounding marine environment features gentle waves and a distant horizon, making this supernatural feat even more striking against the realistic backdrop."* (**2nd**) *"A photorealistic timelapse captures the surreal fusion of a sunflower and rose, creating a striking hybrid bloom. The distinctive yellow petals of the sunflower gradually interweave with the deep red rose petals, while maintaining the recognizable features of both flowers. The transformation occurs against a soft, natural background in natural daylight."*

- ***Right***: (**1st**) *"The smooth black river flows next to the tall green trees. The dark, glossy river creates a stark contrast against the lush, verdant foliage."* (**2nd**) *"A large mountain stands tall in the background, its rugged surface covered in snow and ice. In the foreground, a small tree with delicate leaves and a slender trunk stands proudly, its branches reaching towards the sky."* (**3rd**) *"A wise and ancient tree, symbolic of Russian landscapes, particularly those vast and dense forests of Siberia, stands tall against a backdrop of snowy branches and lush greenery, with sunlight filtering through the canopy, creating a serene and culturally rich atmosphere."* (**4th**) *"A portrait in the style of Leonardo da Vinci, featuring an elderly man with a thoughtful expression, dressed in Renaissance garb, sitting in a dimly lit study adorned with classical paintings and ancient books, with a soft golden light illuminating his weathered face and hands."* (**5th**) *"A rustic peasant scene featuring a thatched-roof cottage with weathered brown walls and red clay tiles, surrounded by vibrant sunflowers and lush green fields under a golden late-afternoon sky, painted in the swirling, expressive style of early Van Gogh."* (**6th**) *"A lone cherry blossom tree stands against a backdrop of autumn's warm hues, its branches heavy with soft pink blossoms set against a golden yellow sky at sunset."*

## B.5. More Details of User Study

To quantitatively evaluate the cognitive alignment between LatentMorph's invocation policy and human intuition, as shown in Figure 6 (*Right*), we conduct a controlled user study.

We first invite 10 human evaluators with experience in visual generation. The evaluation set consists of 20 complex prompts randomly sampled from the WISE and T2I-CompBench benchmarks, ensuring a diverse range of reasoning difficulties. An interactive interface is then developed to simulate the autoregressive generation stream. The procedure is defined as follows: ❶ *Streaming Simulation*: For each prompt, the image is generated token-by-token. To match the model's internal monitoring granularity, the process pauses at every check interval of $w = 64$ tokens. ❷ *Blind Judgment*: At each pause, the evaluator is presented with the intermediate image decoded from the current partial tokens alongside the text prompt. The evaluator is then asked a binary question: *"Given the current progress and the prompt, is it necessary to pause and reason to correct potential errors or refine details?"* ❸ *Decision Collection*: The human decisions (Invoke/Continue) are recorded without revealing the model's actual choices to avoid bias. The Human Alignment score is calculated as the percentage of intervals where the model's action ($a_k \in \{$REASON, CONTINUE$\}$) matches the majority vote of the human evaluators.

## C. More Results & Sensitivity Analysis

### C.1. Results on T2I-CompBench++

To provide a more granular evaluation of compositional generation capabilities, we extend our evaluation to T2I-CompBench++, as shown in Table 3. Consistent with the findings in the main context, LatentMorph demonstrates superior performance across all evaluated dimensions. While recent external reason-while-generation baselines, *e.g.*, TwiG-RL, have made strides in counting capabilities, *i.e.*, Numeracy, LatentMorph pushes the boundary further. We observe consistent gains in additional complex tasks like 3D-Spatial and Numeracy beyond T2I-CompBench. We attribute this to the fact that while explicit text can convey basic quantities, the precise spatial arrangement and disentanglement of multiple objects are better modulated through continuous latent guidance, enabling LatentMorph to resolve intricate spatial-numerical constraints more effectively.

### C.2. More Analysis of Condensers

In this section, we investigate the impact of visual memory capacity on generation performance by varying the latent token lengths for both the short-term $\mathcal{C}_{\text{short}}$ and long-term $\mathcal{C}_{\text{long}}$ condensers. The results are shown in Figure 7.

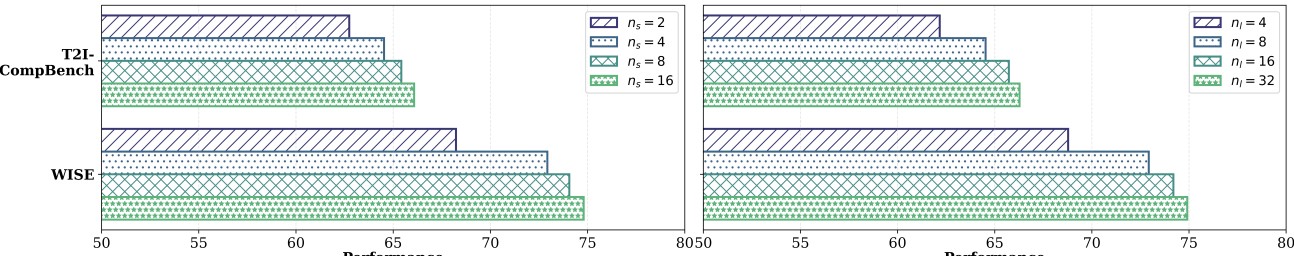

*Figure 7.* Ablation study of latent memory of $\mathcal{C}_{\text{short}}$ (***Left***) and $\mathcal{C}_{\text{long}}$ (***Right***).

**Length of Short-Term Memory.** We examine the sensitivity of the short-term condenser by scaling the memory length $n_s \in \{2, 4, 8, 16\}$. As shown in Figure 7 (*Left*), we observe a consistent improvement in generation quality as the memory token length increases. This suggests that a larger local memory buffer $n_s$ enhances the model's ability to capture fine-grained local dynamics, thereby providing the invoker $\mathcal{I}_{\text{incoker}}$ with more discriminative features for robust decision-making.

**Length of Long-Term Memory.** Similarly, we also ablate the long-term memory length $n_l \in \{4, 8, 16, 32\}$. As demonstrated in Figure 7 (*Right*), expanding the global memory size yields progressive performance gains. We attribute this to the increased representational capacity of the visual memory. A larger $n_l$ effectively alleviates the information bottleneck, enabling the reasoning core $\text{UMM}_u$ to access a more granular and comprehensive summary of the entire generation history.

### C.3. More Analysis of Invoker

In this section, we provide a granular analysis of the invoker $\mathcal{I}_{\text{invoker}}$, validating its design choices regarding invocation timing, monitoring granularity, and state representation.

**Invocation Timestep.** We first evaluate the necessity of our learned adaptive policy against heuristic baselines. Table 4 compares our method with random injection *i.e.*, with probabilities $p \in \{0.3, 0.5, 0.7\}$, and fixed schedules *i.e.*, injecting once at the midpoint or twice at $1/3$ and $2/3$ intervals. The results demonstrate that our adaptive strategy consistently outperforms both stochastic and rigid interventions. We attribute this to context awareness: while fixed strategies may intervene during trivial generation phases (*e.g.*, wasting computation) or miss critical turning points, our RL-trained $\mathcal{I}_{\text{invoker}}$ learns to trigger reasoning precisely when semantic drift or high uncertainty is detected.

*Table 4.* Ablation study of invocation strategies.

| Invocation | Color ↑ | Shape ↑ | Texture ↑ | Spatial ↑ | Non-Spatial ↑ | Comeplex ↑ | Overall ↑ |
|---|---|---|---|---|---|---|---|
| Random ($p = 0.3$) | 78.72 | 63.71 | 76.00 | 45.73 | 37.67 | 51.73 | 58.93 |
| Random ($p = 0.5$) | 78.56 | 64.83 | 76.07 | 48.00 | 37.93 | 51.87 | 59.54 |
| Random ($p = 0.7$) | 77.42 | 65.63 | 76.07 | 49.60 | 39.20 | 51.67 | 59.93 |
| Fixed (1) | 78.28 | 65.29 | 73.78 | 46.73 | 37.80 | 51.20 | 58.85 |
| Fixed (2) | 77.61 | 63.29 | 73.33 | 46.27 | 38.33 | 50.87 | 58.28 |
| **Ours** | **84.04** | **69.46** | **79.89** | **50.93** | **39.56** | **63.60** | **64.53** |

**Window Size of Checking.** We examine the impact of the monitoring window size $w$ in Table 5. We observe that performance is sensitive to the temporal resolution of monitoring. A small window $w = 32$ tends to capture local noise in token prediction, leading to erratic decision-making, while an overly large window $w = 128$ suffers from lag, failing to detect rapid compositional shifts in time. Our default setting of $w = 64$ offers an optimal trade-off, providing a stable yet responsive signal for the invoker.

*Table 5.* Ablation study of check interval.

| Window Size | Color ↑ | Shape ↑ | Texture ↑ | Spatial ↑ | Non-Spatial ↑ | Comeplex ↑ | Overall ↑ |
|---|---|---|---|---|---|---|---|
| $w = 32$ | 81.67 | 66.47 | 77.56 | 49.40 | 38.06 | 57.00 | 61.69 |
| $w = 64$ (**Ours**) | **84.04** | **69.46** | **79.89** | **50.93** | 39.56 | **63.60** | **64.53** |
| $w = 128$ | 82.01 | 66.10 | 76.89 | 49.68 | **40.05** | 60.45 | 62.53 |

**Multi-dimensional Signals.** We further conduct an ablation study on the components of the state vector $s_i$ in Table 6. The results indicate that all four signal dimensions, *i.e.*, semantic consistency $c_i$, uncertainty $u_i$, temporal dynamics $\Delta c_i$, and stability $v_i$, are essential for robust performance. Notably, removing uncertainty $u_i$ or semantic consistency $c_i$ leads to the most significant performance drops, confirming that model confidence and alignment scores are the primary indicators for determining the necessity of latent reasoning.

*Table 6.* Ablation study of state vector $s_i$.

| State Signal | Color ↑ | Shape ↑ | Texture ↑ | Spatial ↑ | Non-Spatial ↑ | Comeplex ↑ | Overall ↑ |
|---|---|---|---|---|---|---|---|
| w/o Semantic Consistency $c_i$ | 79.41 | 65.90 | 78.13 | 48.37 | 39.20 | 57.93 | 61.49 |
| w/o Uncertainty $u_i$ | 79.10 | 66.32 | 78.61 | 49.70 | 38.32 | 59.55 | 61.93 |
| w/o Temporal Dynamics $\Delta c_i$ | 82.78 | 68.77 | 79.14 | 50.57 | 38.30 | 61.39 | 63.49 |
| w/o Stability $v_i$ | 83.01 | 68.24 | 78.99 | 50.21 | 39.19 | 62.00 | 63.61 |
| **Ours** | **84.04** | **69.46** | **79.89** | **50.93** | **39.56** | **63.60** | **64.53** |

## C.4. More Analysis of Translator

In this section, we dissect the composition of the control signals generated by the latent translator $\mathcal{T}_{\text{trans}}$. Recall that our design in Section §4.3 fuses the latent thought $\mathbf{z}$ with two critical context signals, *i.e.*, the long-term visual memory $\mathbf{m}^{(l)}$ and the original prompt embedding $\mathbf{p}$. We evaluate the contribution of each component in Table 7.

**Control Signals.** The ablation results demonstrate that relying solely on latent thoughts $\mathbf{z}$ is insufficient for precise control, and both visual context and textual grounding are indispensable.

❶ **Impact of Visual Memory:** Removing the long-term visual memory $\mathbf{m}^{(l)}$ leads to a marked decline in performance. This indicates that $\mathbf{m}^{(l)}$ provides essential historical context, allowing the translator to align abstract reasoning with the visual content generated so far. Without it, the control signals risk disrupting global visual consistency.

❷ **Impact of Prompt Embedding:** Similarly, omitting the prompt embedding $\mathbf{p}$ results in inferior alignment scores. We attribute this to the role of $\mathbf{p}$ as a semantic anchor, which ensures that the translated control guidance remains strictly grounded in the user's original intent, preventing the reasoning process from drifting into unconstrained generation.

*Table 7.* Ablation study of control signals.

| Contol Signal | Color ↑ | Shape ↑ | Texture ↑ | Spatial ↑ | Non-Spatial ↑ | Comeplex ↑ | Overall ↑ |
|---|---|---|---|---|---|---|---|
| w/o Visual Memory $\mathbf{m}^{(l)}$ | 82.52 | 66.12 | 76.90 | 48.00 | 38.39 | 61.98 | 62.40 |
| w/o Prompt Embedding $\mathbf{p}$ | 82.20 | 67.42 | 77.01 | 47.21 | 38.10 | 62.10 | 62.34 |
| **Ours** | **84.04** | **69.46** | **79.89** | **50.93** | **39.56** | **63.60** | **64.53** |

## C.5. More Analysis of Shaper

Finally, in this section, we validate the efficacy of the latent shaper $\mathcal{S}_{\text{shaper}}$ by investigating how reasoning signals are integrated into the generation stream. We compare our KV-cache injection strategy against a baseline variant "w/o $\mathcal{S}_{\text{shaper}}$", where the translated latent control signals directly replace the original prompt embeddings $\mathbf{p}$ instead of being appended as additional context. The results are shown in Table 8.

**Control Injection.** The experimental results indicate that the direct replacement strategy, *i.e.*, w/o $\mathcal{S}_{\text{shaper}}$, yields suboptimal performance compared to our additive injection approach. We attribute this to the fact that overwriting the prompt embedding is destructive, as it severs the model's link to the global generation objective, causing it to lose track of the initial user

instruction while focusing on local refinements. In contrast, our $\mathcal{S}_{\text{shaper}}$ injects control tokens $\mathbf{E}_{\text{ctrl}}$ into the KV cache, acting as a soft modulation mechanism. This design preserves the integrity of the original prompt while seamlessly steering the attention dynamics based on latent reasoning cues, ensuring that refinement does not come at the cost of global semantic fidelity.

*Table 8.* Ablation study of control injection.

| Contol Injection | Color ↑ | Shape ↑ | Texture ↑ | Spatial ↑ | Non-Spatial ↑ | Comeplex ↑ | Overall ↑ |
|---|---|---|---|---|---|---|---|
| w/o $\mathcal{S}_{\text{shaper}}$ | 79.42 | 65.42 | 77.21 | 47.32 | 37.90 | 60.21 | 61.25 |
| **Ours** | **84.04** | **69.46** | **79.89** | **50.93** | **39.56** | **63.60** | **64.53** |

## D. Exhibition Board

We provide more comparison results here in Figure 8 and 9.

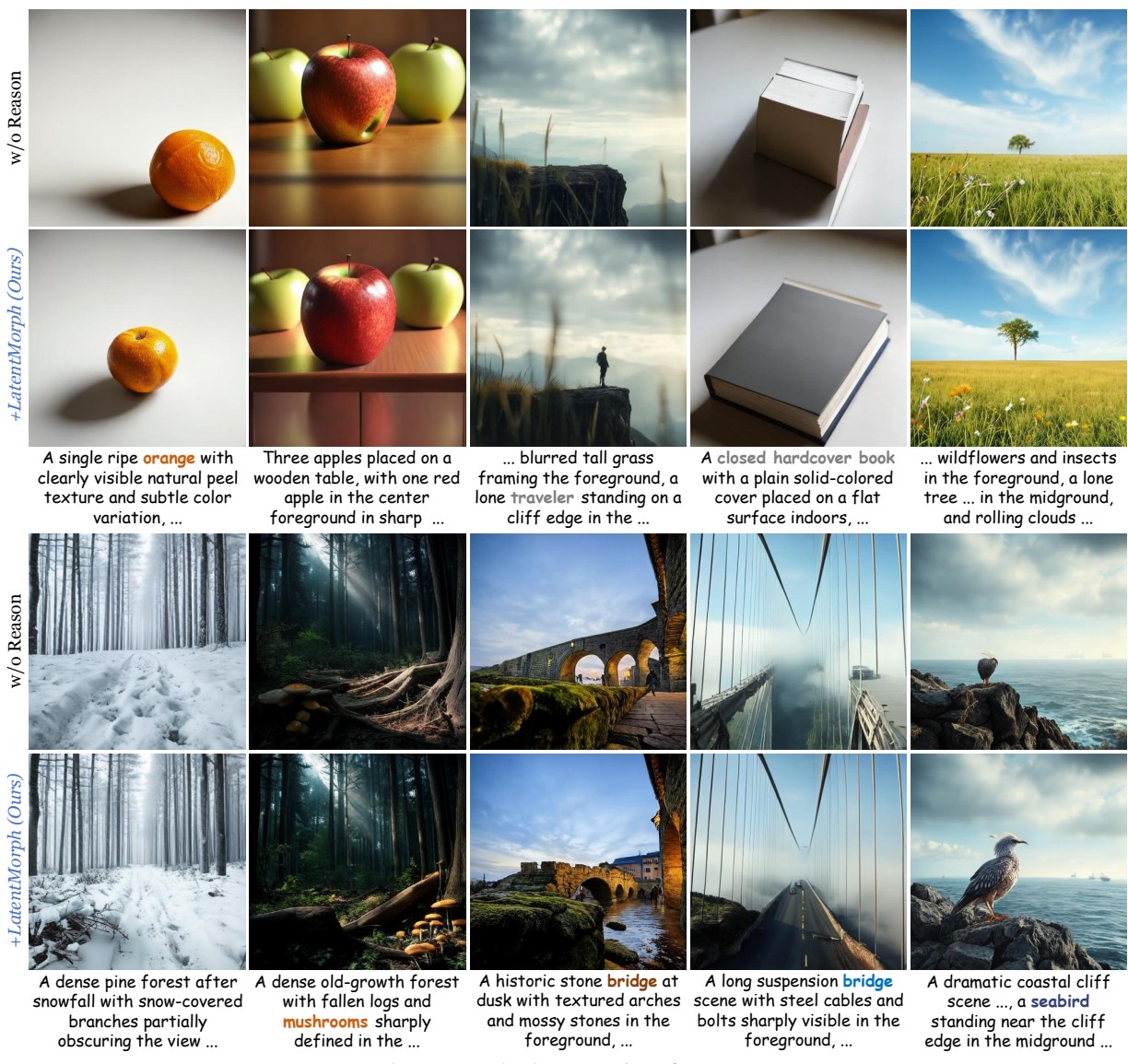

*Figure 8.* More results demonstration of LatentMorph.

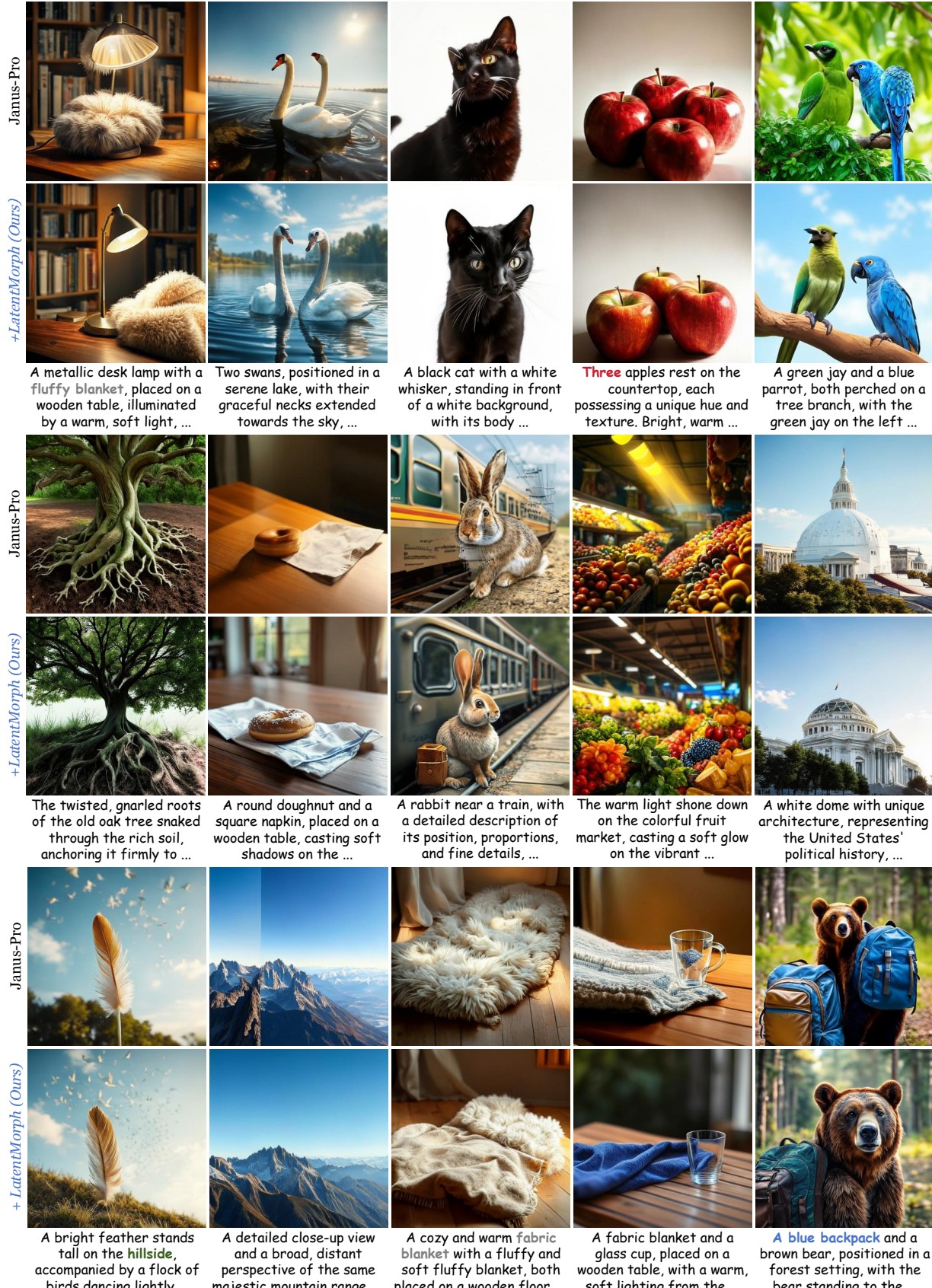

*Figure 9.* More results demonstration of LatentMorph.

