# OpenReview forum: "Show, Don't Tell: Morphing Latent Reasoning into Image Generation"
_ICML.cc/2026/Conference — ICML 2026 regular_

### Official Review · Reviewer_n4Na · 2026-02-21

**Soundness:** 2
**Presentation:** 2
**Significance:** 3
**Originality:** 3
**Overall Recommendation:** 4
**Confidence:** 3

**Summary:**

The paper proposes LatentMorph, a reasoning-augmented text-to-image framework that moves “reasoning” away from explicit textual CoT and into latent-space interventions woven into the autoregressive generation process. It builds a closed loop where intermediate generation states are compressed into visual memories, converted into latent guidance, injected into the generator through KV-cache control, and triggered only when needed via an RL-trained invoker. Experiments on a Janus-Pro backbone report consistent quality gains across multiple benchmarks alongside reduced overhead from avoiding explicit decode–re-encode reasoning loops.

**Compliance With Llm Reviewing Policy:**

Affirmed.

**Ethical Review Concerns:**

No specific ethical concerns are raised in the paper.

**Key Questions For Authors:**

It would help to quantify how much of the improvement truly depends on the latent thought representation rather than memory signals or generic control tokens. I would also like to understand how stable and transferable the RL-trained invoker is across different prompt distributions and model backbones, and whether the reported efficiency comparisons include the full cost of the understanding branch and control injection under matched decoding settings. Finally, the human-agreement results are interesting, but they need more methodological detail (annotator agreement, task breakdown, and labeling protocol) to be interpretable.

**Limitations:**

The paper acknowledges both benefits and risks and notes that the approach inherits the backbone model’s constraints.

**Strengths And Weaknesses:**

Strengths
The authors examine a notable theme: making reasoning native to image generation by keeping it implicit, continuous, and intertwined with autoregressive decoding rather than externalizing it as text. The core mechanism is conceptually coherent and practically attractive, especially the KV-cache–based control injection that can steer later token prediction without repeatedly decoding intermediate reasoning into text or images. The decision to treat when to reason as a learnable policy also strengthens the narrative, since it matches the intuition that difficult prompts benefit from intervention while easy ones do not.

Weaknesses
The method combines several modules and training stages, including RL for the invoker and reward-driven optimization, which raises the risk that results depend on implementation details and may be harder to reproduce faithfully. The paper would also benefit from sharper evidence that the gains arise from “latent reasoning” itself rather than from improved conditioning, memory summarization, or control-token injection that functions similarly to a strong adapter. Finally, efficiency comparisons and claims of model-agnostic generality are plausible but not fully airtight as presented; they would be stronger with stricter apples-to-apples runtime protocols and broader validation on additional backbones or domains.

---

> ### Author Rebuttal · Authors · 2026-03-30
>
> We would like to express our deepest respect for your meticulous review! We are also encouraged that the reviewers recognized the importance of the problem, our empirical evaluation, and the coherence of the proposed framework.
>
> In response to your efforts, we have carefully prepared a point-by-point reply:
>
> ---
> > `Weakness 1`: Pipeline complexity and reproducibility.
>
> We agree reproducibility is important and designed LatentMorph accordingly:
>
> - **Stable training:** LatentMorph updates only lightweight added modules (**<8.5%** parameters) plus a backbone LoRA, rather than full-parameter RLHF. This constrained optimization space, together with robust GRPO frameworks such as DanceGRPO and Janus-Pro-R1 [1], yields stable training without heavy hyperparameter tuning.
>
> - **Reproducibility:** We have already provided the complete anonymous codebase, including exact hyperparameters, reward definitions, and training scripts for both the *two-stage* SFT and RL pipeline.
>
> ---
> > `Weakness 2 & Question 1`: Nature and quantification of latent reasoning.
>
> We deeply appreciate this insightful critique. The fundamental motivation of our work is to overcome the bottlenecks of explicit text-based reasoning by conducting intermediate computations **purely in the continuous latent space**. We respectfully clarify that the empirical gains stem precisely from latent reasoning, not passive conditioning or memory summarization, supported by two results:
>
> - **What is represented (`Figure 5`):** In the `LatentMorph w/o latent` ablation in `Figure 5`, forcing latent thoughts into discrete text drops WISE/IPV-Txt from **72.98/49.42** to **64.18/42.78**, a **6.4–8.8** point decrease. This quantifies the loss from verbalizing high-dimensional intermediate states.
>
> - **When to reason (`Table 4`):** Replacing the learned Invoker with random or fixed-step injection significantly degrades performance, showing that the gain comes from context-aware step-level reasoning decisions rather than passive feature enhancement.
>
> ---
> > `Weakness 3 & Question 2`: Generality and full-cost efficiency accounting.
>
> We appreciate the push for stricter empirical validation. We clarify the stability and efficiency of LatentMorph as follows:
>
> - **Generality:** We further evaluate LatentMorph on two entirely distinct architectures: **Tar** [2] (a *hybrid AR-Diffusion* UMM) and **Qwen2.5-VL+LlamaGen** (a *decoupled* external-loop setup). As shown in **Table A**, LatentMorph consistently outperforms vanilla and explicit 3-step (TwiG's setting) baselines.
>
>     *Table A: Generalization across architectures.*
>     |Paradigm|Model|Reasoning Strategy|GenEval|
>     |-|-|-|-|
>     |Internal-Loop|Tar|Gen.-Only (Vanilla)|0.76|
>     |||Explicit 3-Step|0.79|
>     |||**LatentMorph (Ours)**|**0.86**|
>     |External-Loop|Qwen2.5-VL+LlamaGen|Gen.-Only (LlamaGen)|0.32|
>     |||Explicit 3-Step|0.36|
>     |||**LatentMorph (Ours)**|**0.42**|
>
> - **Efficiency:** All reported efficiency metrics (*e.g.*, **44% speedup**, **51% token reduction**) use a *matched end-to-end decoding protocol* on a single H200, including the understanding branch, latent computation, and control injection. The advantage comes from reasoning in latent space rather than repeatedly decoding/encoding text.
>
> ---
> > `Question 3`: Methodological details of the user study.
>
> To ensure a fair and interpretable comparison of human-cognitive alignment, we followed a strictly controlled protocol for both LatentMorph and the TwiG baseline:
>
> - **Setup:** We randomly sampled 20 complex prompts from WISE and T2I-CompBench. 10 experienced evaluators performed a **blind binary task** at every $w=64$ tokens, deciding whether a reasoning pause was necessary based on the current partially decoded image.
>
> - **Metric:** We calculate the **Human Alignment Score** as the percentage of checkpoints where the model's action matches the human majority vote. (1) LatentMorph: Its learned Invoker dynamically decides at each 64-token interval. (2) TwiG Baseline: Since TwiG uses a fixed 3-step invocation (at $0, 1/3, 2/3$ of total tokens), we mapped its two mid-generation invocations to the nearest 64-token checkpoints for a direct comparison.
>
> The stark gap (**71.83% vs. 18.67%**) proves that human-like reasoning is inherently **event-driven and dynamic**, not periodic. LatentMorph better learns this "when-to-reason" intuition, whereas fixed-step methods like TwiG largely fail to trigger at cognitively meaningful moments.
>
> ---
> Thank you again for your detailed feedback. We hope this rebuttal clarifies the points raised and addresses your concerns. We are happy to continue discussing ways to improve the paper.
>
> ---
> [1] Janus-Pro-R1: Advancing Collaborative Visual Comprehension and Generation via Reinforcement Learning, NeurIPS’25
>
> [2] Vision as a Dialect: Unifying Visual Understanding and Generation via Text-Aligned Representations, NeurIPS’25

---

> > ### Author Rebuttal · Reviewer_n4Na · 2026-04-03
> >
> > The rebuttal clarifies that the gains come from latent reasoning rather than generic conditioning, and adds useful details on generality, efficiency, and the human study protocol. While annotator agreement could still be reported more clearly.

---

> > > ### Author Response · Authors · 2026-04-03
> > >
> > > Dear Reviewer n4Na,
> > >
> > > Thank you for your constructive follow-up and for **recognizing our clarifications regarding latent reasoning, generality, and efficiency**.
> > >
> > > We completely agree that the human study details should be reported more transparently. To directly address your request for a clearer **task breakdown** and **annotator agreement**, we provide the exact evaluation protocol:
> > >
> > > ---
> > >
> > > **1. Task Breakdown & Metric Definition:**
> > > Rather than a global accuracy score, we calculated the **Human Endorsement Rate (Expected Precision)** of the model's actual interventions. This strictly measures: *When the model chooses to invoke reasoning, what percentage of human annotators agree that a pause was visually necessary?*
> > >
> > > * **The Process:** We monitored the sequence at 7 candidate checkpoints per image (excluding start/end). For a fair comparison, both LatentMorph and TwiG were evaluated on the same total number of intervention points across our 20 test images.
> > >
> > > * **The Validation Logic:** At each evaluated intervention point, the model's decision is validated against 10 independent blind votes from our expert annotators. This ensures the endorsement rate rigorously captures the variance in human cognition, explicitly penalizing fixed-step baselines that trigger reasoning at visually arbitrary moments.
> > >
> > > ---
> > >
> > > **2. Annotator Agreement:**
> > > To validate the reliability of the human ground truth, we further measure inter-rater agreement across all evaluated points here:
> > >
> > > * **Average Consensus:** An average of ~84% of annotators aligned with the final majority vote.
> > >
> > > * **Fleiss' Kappa ($\kappa$):** We achieved a score of $\kappa = 0.67$, indicating substantial agreement in statistical literature.
> > >
> > > ---
> > >
> > > **3. Commitment to Transparency:**
> > > To ensure full reproducibility and clarity on these metrics, *we will open-source our user study calculation scripts* along with our code and model checkpoints upon publication.
> > >
> > > ---
> > >
> > > We will explicitly add this breakdown to the revised `Appendix B.5`. Thank you again for helping us elevate the rigorousness of our reporting!
> > >
> > > Best regards,
> > >
> > > Authors

---

### Official Review · Reviewer_QxiX · 2026-03-11

**Soundness:** 2
**Presentation:** 3
**Significance:** 2
**Originality:** 2
**Overall Recommendation:** 4
**Confidence:** 5

**Summary:**

This paper proposes framework LatentMorph to improve the T2I process. It replaces the traditional explicit textual reasoning with implicit latent reasoning to avoid information loss and inefficiency. The proposed framework contains four lightweight components: a condenser to summarize generation states into visual memory, a translator to convert latent thoughts, a shaper to dynamically steer token predictions, and an invoker to adaptively determine when to trigger the reasoning. Extensive experiments on multiple benchmarks show that LatentMorph can enhance the base model Janus-Pro and reduce the inference time.

**Compliance With Llm Reviewing Policy:**

Affirmed.

**Final Justification:**

I appreciate the authors' effort and will increase my score. I expect the authors to include all the refinements they promised in the revised version. Meanwhile, I still have concerns about the framework's complexity. Its design is not quite in line with the current trend of simple and unified MLLMs. I respect the AC's final decision.

**Key Questions For Authors:**

1. The latent shaper injects control tokens directly into the KV cache of the generator to modify subsequent predictions. How does this dynamic injection operation affect the absolute positional embeddings and relative distances of the original token sequence during the autoregressive generation?

2. The paper claims that LatentMorph can be applied to external-loop paradigms. Do you have any quantitative experimental results that apply this framework to a pure diffusion model or a non-UMM generator to verify this claim?

**Limitations:**

Yes

**Strengths And Weaknesses:**

Strengths:

1. The motivation of migrating the reasoning process from the discrete textual space into the continuous latent space makes good sense.

2. The authors conduct rich experiments on diverse benchmarks, including GenEval, T2I-CompBench, and WISE.


Weaknesses:

1. The framework design is overly complex and the story is somewhat overclaimed. Given the heavy training pipeline and framework, the performance gains are not particularly obvious. Compared to other baselines, although the model uses latent reasoning to save inference time, the overall training cost has significantly increased.

2. The method relies heavily on the UMM architecture and is only tested on Janus-Pro. The authors claim in the appendix that it works with decoupled external-loop generators, but without real empirical results to back this up, the generalization ability of the framework is questionable.

---

> ### Author Rebuttal · Authors · 2026-03-30
>
> Thank you for your thoughtful and constructive feedback! We are also encouraged that the reviewers recognized the importance of the problem, our empirical evaluation, and the coherence of the proposed framework.
>
> Based on your questions, we give point-by-point responses to your comments:
>
> ---
> > `Weakness 1`: Training complexity, cost, and performance gains.
>
> We respectfully clarify a likely misunderstanding regarding our training cost, which may stem from our transparent reporting of an 8x H200 GPU cluster. This hardware was used strictly to accelerate experimental turnaround time, **not** out of necessity.
>
> - **Minimal Training Cost:** LatentMorph is lightweight: the new modules add **<8.5%** parameters, and training is **two-stage** with only 20k **standard** text-image SFT samples, without process-level supervision as in explicit reasoning methods (*e.g.*, 50k annotated data in TwiG). To further prove this, we also train our method with a **strictly frozen backbone**, confirming that it remains effective even under constrained compute (**Table A**).
>
>     *Table A: Parameter overhead and frozen backbone ablation*
>     |Setting|Backbone|GenEval|T2I-CompBench|
>     |-|-|-|-|
>     |Vanilla|N/A|0.80|39.21|
>     |SFT|Train w/ LoRA|0.79|39.43|
>     |GRPO|Train w/ LoRA|0.82|41.61|
>     |**LatentMorph w/ Frozen Backbone**|Frozen|0.94|61.59|
>     |**LatentMorph (Default)**|Train w/ LoRA|**0.96**|**64.53**|
>
> - **Substantial & Efficient Gains:** We respectfully disagree that the gains are not obvious. LatentMorph achieves a massive **+25.32 absolute improvement** on T2I-CompBench over the base model. Crucially, it achieves this while *reducing* inference time by **44%** and token consumption by **51%** compared to explicit reasoning baselines (**Table B**).
>
>     *Table B: Inference efficiency on T2I-CompBench (`Figure 6`).*
>     |Method|Paradigm|Inference Time (s)|Token Consumption (tokens)|Performance|
>     |-|-|-|-|-|
>     |Vanilla|Gen.-Only|1.3e+04|1.0e+06|39.21|
>     |MILR|Iterative Search|1.9e+05|2.2e+07|53.25|
>     |TwiG-ZS|Explicit Reasoning|2.6e+04|2.3e+06|46.75|
>     |**LatentMorph (Ours)**|Implicit Reasoning|**1.4e+04**|**1.1e+06**|**64.53**|
>
> ---
> > `Weakness 2 & Question 2`: Architectural reliance and generalization claims.
>
> We agree that empirical validation is essential to support our claims of generalization. To directly address your concern, we have conducted new experiments on the exact decoupled external-loop paradigm you highlighted, alongside an entirely different internal-loop model:
>
> - **External-Loop (Decoupled Generators):** We apply LatentMorph to a decoupled VLM + Generator setup (**Qwen2.5-VL + LlamaGen**), proving that our framework can successfully act as a differentiable neural interface between completely separate models.
>
> - **Internal-Loop (Hybrid AR-Diffusion UMM):** We also adapt LatentMorph to **Tar** [1], a recently proposed UMM that employs a hybrid AR-Diffusion decoding pipeline, vastly different from our base Janus-Pro.
>
> As shown in **Table C**, LatentMorph consistently outperforms both the vanilla backbones and explicit reasoning (TwiG's setting) baselines across these architectures.
>
> *Table C: Generalization across architectures.*
> |Paradigm|Model|Reasoning Strategy|GenEval|
> |-|-|-|-|
> |Internal-Loop|Tar|Gen.-Only (Vanilla)|0.76|
> |||Explicit 3-Step|0.79|
> |||**LatentMorph (Ours)**|**0.86**|
> |External-Loop|Qwen2.5-VL+LlamaGen|Gen.-Only (LlamaGen)|0.32|
> |||Explicit 3-Step|0.36|
> |||**LatentMorph (Ours)**|**0.42**|
>
> ---
> > `Question 1`: Impact of KV cache injection on positional embeddings.
>
> We appreciate this detailed technical question. As mathematically formalized in `Appendix A.3.2`, our injection mechanism is explicitly designed to preserve the sequence's original positional integrity:
>
> - **No Positional Shift:** When injecting control signals ($K_{\text{ctrl}}, V_{\text{ctrl}}$) into the KV cache at step $k$, they act solely as a "virtual context". Crucially, we **do not increment the absolute positional indices** for any subsequent tokens ($x_{t>k}$).
>
> - **Preserved Relative Distances:** Because the absolute IDs of the main sequence remain contiguous, the internal relative distances between all original tokens are completely unaltered. This ensures LatentMorph remains strictly compatible with Rotary Positional Embeddings (RoPE) without disrupting the base model's spatial dynamics.
>
> ---
> We appreciate your insightful comments again and hope this rebuttal addresses your questions. Please feel free to reach out with further suggestions to improve clarity.
>
> ---
> [1] Vision as a Dialect: Unifying Visual Understanding and Generation via Text-Aligned Representations, NeurIPS’25

---

> > ### Author Rebuttal · Reviewer_QxiX · 2026-04-03
> >
> > Thank you for the detailed rebuttal. It addresses my questions on the KV-cache injection mechanism, and also partially addresses my concern on generalization beyond Janus-Pro.
> >
> > However, my main concern still remains on the overall training cost and framework complexity. The rebuttal clarifies the parameter overhead and supervision requirement, but my main question is the **actual cost of the two-stage SFT + RL pipeline, including the training of the condenser, invoker, translator, and shaper.** My understanding is that these components all need additional training, and I am still not fully convinced that the added complexity is justified.
> >
> > More generally, while I agree this is a relatively complete paper with sufficient experiments and ablations, I still see it mainly as an engineering-heavy mechanism paper. My concern is not mainly whether the method can work, but whether the overall complexity is worthwhile, and whether the paper provides enough community-level insight.
> >
> > I also still have concern that some claims are somewhat overstated. For example, the paper raises empirical improvements to the claim that latent reasoning preserves 'ineffable semantics.' In my opinion, the current evidence, including the ablations and the differential heatmap, can show that latent reasoning changes the model's visual focus, but it cannot directly prove that these regions correspond to 'ineffable semantics,' or that explicit reasoning necessarily loses such semantics. Similarly, results mainly validated on Janus-Pro are presented as a broader paradigm. Even with the added Qwen + LlamaGen results, the method still appears largely tied to autoregressive generation.
> >
> > Overall, I see the contribution more as turning a reasonable idea into a complete system and showing that it can work on benchmarks, rather than providing a fundamentally new insight. That said, I appreciate the additional experiments and will respect the AC's final decision.

---

> > > ### Author Response · Authors · 2026-04-03
> > >
> > > Dear Reviewer QxiX,
> > >
> > > Thank you for engaging in this discussion and for your candid, constructive feedback. We deeply appreciate your recognition that our idea is reasonable and works well. Your points regarding the training cost and the phrasing of our claims are highly pertinent. We would like to clarify these final aspects:
> > >
> > > ---
> > >
> > > > `1. The Actual Training Cost.`
> > >
> > > To be fully transparent, the "heavy pipeline" perception might be a misunderstanding. The entire two-stage training is extremely lightweight and fast, taking **only ~7 hours in total**:
> > >
> > > * **Stage 1 (SFT):** Trains *only* the long condenser and translator. It takes just **~2 hours** to process 20k samples (*total batch size 16*).
> > >
> > > * **Stage 2 (RL):** Trains *only* the short condenser and invoker. It takes just **~5 hours** for ~1,500 prompts. Specifically, using GRPO, we employ a *prompt batch size of 1 with 8 rollouts*, and we observe that the policy remarkably **converges in only ~500 steps**.
> > >
> > > Given the +25.32 absolute gain, this ~7-hour localized training overhead is negligible compared to the massive cost of standard visual pre-training.
> > >
> > > ---
> > >
> > > > `2. Clarification on "Ineffable Semantics".`
> > >
> > > You raise an excellent scientific point. We completely agree that while our heatmaps and ablations show a clear shift in visual focus and improved performance, they may not strictly prove the preservation of "ineffable semantics". We accept this critique and will tone down our claims. In the revised manuscript, we will replace terms like "ineffable semantics" with more precise, empirically supported descriptions, such as **"fine-grained, high-dimensional visual features that are difficult to fully compress into discrete text tokens"**. Thank you for helping us make the paper scientifically tighter.
> > >
> > > ---
> > >
> > > > `3. Community-Level Insight and Generalization Beyond Pure AR.`
> > >
> > > Regarding the concern that our method is tied to AR generation and lacks broader insight: our core **community-level insight** is demonstrating that *shifting from explicit discrete tokens to implicit continuous states offers a highly effective and efficient new paradigm for reasoning-enhanced visual generation.*.
> > >
> > > To prove this is not merely an AR-specific strategy, we respectfully draw your attention to the **Tar** experiment in our newly added **Table C**. Tar employs a **hybrid AR-Diffusion** pipeline with a continuous diffusion image head. The fact that LatentMorph successfully generalizes to Tar provides concrete evidence that our framework extends beyond pure AR paradigms.
> > >
> > > ---
> > > We sincerely thank you for your time and constructive feedback, which have helped us make the paper tighter and more precise.
> > >
> > > Best regards,
> > >
> > > Authors

---

### Official Review · Reviewer_WPuU · 2026-03-12

**Soundness:** 3
**Presentation:** 4
**Significance:** 3
**Originality:** 3
**Overall Recommendation:** 4
**Confidence:** 4

**Summary:**

LatentMorph integrates implicit latent reasoning into autoregressive text-to-image generation.
It introduces condensers for short- and long-term visual memory, a translator and shaper to inject latent thoughts into the generator KV cache, and an RL-trained invoker to decide when to reason.
The method improves Janus-Pro on GenEval and T2I-CompBench, outperforms explicit reasoning baselines on WISE and IPV-Txt, and reduces inference time and token usage.

**Compliance With Llm Reviewing Policy:**

Affirmed.

**Key Questions For Authors:**

1) How does LatentMorph perform on other UMMs or diffusion-based generators?
2) What is the added runtime overhead of condensers and invoker relative to the base model?
3) How is the RL reward constructed and tuned, and how stable is training?
4) Are there failure cases where the invoker over-triggers or under-triggers reasoning?
5) Does latent reasoning introduce new safety or content drift issues?

**Limitations:**

Not fully; please discuss dependence on autoregressive UMMs with hidden-state access, training compute, and potential safety risks.

**Strengths And Weaknesses:**

Strengths:
1) Addresses key inefficiencies of explicit reasoning with a coherent latent alternative.
2) Adaptive invocation and memory design are well motivated, with extensive ablations.
3) Demonstrates gains in both quality and efficiency across multiple benchmarks.

Weaknesses:
1) Many moving parts; overall complexity and training cost are high.
2) Generalization to other generators and diffusion models is not shown.
3) RL reward definition and stability are not fully specified; limited qualitative failure analysis.

---

> ### Author Rebuttal · Authors · 2026-03-30
>
> We sincerely thank you for the thoughtful and constructive reviews of our manuscript! We are also encouraged that the reviewers recognized the importance of the problem, our empirical evaluation, and the coherence of the proposed framework.
>
> In response to your efforts, we have carefully prepared a point-by-point reply:
>
> ---
> > `Weakness 1 & Question 2`: Complexity, training cost & runtime overhead.
>
> We respectfully clarify this from two perspectives:
>
> - **Lightweight modules and tractable training:** LatentMorph adds **<8.5%** parameters and is trained in **two stages**. The SFT stage uses only 20k **standard** text-image pairs, without process-level supervision as in explicit reasoning methods (*e.g.*, 50k annotated data in TwiG [1]). It remains effective even with a *frozen backbone*, updating only our lightweight modules.
>
>     *Table A: Parameter overhead and frozen backbone ablation*
>     |Setting|Backbone|GenEval|T2I-CompBench|
>     |-|-|-|-|
>     |Vanilla|N/A|0.80|39.21|
>     |SFT|Trainable w/ LoRA|0.79|39.43|
>     |GRPO|Trainable w/ LoRA|0.82|41.61|
>     |**LatentMorph w/ Frozen Backbone**|Frozen|0.94|61.59|
>     |**LatentMorph (Default)**|Trainable w/ LoRA|**0.96**|**64.53**|
>
> - **Minimal runtime overhead:** Relative to vanilla, LatentMorph adds only **~7.7%** runtime and **10%** token usage, while substantially improving performance (*i.e.*, **+ 25.32** absolute gain); it is also much more efficient than explicit reasoning baselines.
>
>     *Table B: Inference efficiency on T2I-CompBench (`Figure 6`).*
>     |Method|Paradigm|Time (s)|Token Consumption|Performance|
>     |-|-|-|-|-|
>     |Vanilla|Gen.-Only|1.3e+04|1.0e+06|39.21|
>     |MILR|Search|1.9e+05|2.2e+07|53.25|
>     |TwiG-ZS [1]|Explicit Reason.|2.6e+04|2.3e+06|46.75|
>     |**LatentMorph (Ours)**|Implicit Reason.|**1.4e+04**|**1.1e+06**|**64.53**|
>
> ---
> > `Weakness 2 & Question 1`: Generalization to other architectures.
>
> As discussed in `Appendix A.3`, LatentMorph is designed to be model-agnostic with a latent-state interface. Here we further add results on **Tar** [2] (*hybrid AR-diffusion* UMM, internal-loop) and **Qwen2.5-VL+LlamaGen** (*decoupled* external-loop). In both cases, it outperforms the vanilla backbone and an explicit 3-step (TwiG's setting) baseline.
>
> *Table C: Generalization across architectures.*
> |Paradigm|Model|Reasoning Strategy|GenEval|
> |-|-|-|-|
> |Internal|Tar|Gen.-Only (Vanilla)|0.76|
> |||Explicit 3-Step|0.79|
> |||**LatentMorph (Ours)**|**0.86**|
> |External|Qwen2.5-VL+LlamaGen|Gen.-Only (LlamaGen)|0.32|
> |||Explicit 3-Step|0.36|
> |||**LatentMorph (Ours)**|**0.42**|
>
> ---
> > `Weakness 3 & Question 3, 4`: RL reward design, stability & failure cases.
>
> We thank you for requesting further details. We further clarify the RL setup (`Appendix A.1.2`) and failure modes below:
>
> - **Reward:** Weighted sum of CLIP and HPS-v2.1 (following robust DanceGRPO), plus an *adaptive invocation penalty* to discourage trivial over-triggering.
>
> - **Stability:** Built on Janus-Pro-R1 [3] and mainly updates lightweight modules rather than full backbone; in practice, training is stable, with no abnormal collapse or invocation explosion.
>
> - **Failure Cases Analysis:** (1) *Over-triggering* on benign high-frequency textures (*e.g.*, waves, grass), causing minor discontinuities; (2) *Under-triggering* at true compositional bottlenecks, reverting to vanilla AR errors (*e.g.*, wrong counts). *We will add examples in the revision.*
>
> ---
> > `Question 5`: Does latent reasoning introduce new safety or content drift issues?
>
> We humbly clarify the concerns below:
>
> - **Safety:** We do not observe evidence of new safety issues beyond the base model. LatentMorph uses only **standard academic** text-image datasets and largely inherits the underlying generator’s safety behavior.
>
> - **Content Drift:** The *Latent Translator* re-injects the original prompt embedding at every invocation step; `Table 7` in Appendix shows this semantic anchor is important for maintaining prompt alignment.
>
> ---
> > `Limitation`: Please discuss dependence on autoregressive UMMs with hidden-state access, training compute, and potential safety risks.
>
> We fully agree that explicitly discussing these boundaries provides a more holistic view of our work. As suggested, we will add a dedicated "Limitations and Broader Impacts" section in the revised manuscript, summarizing our detailed discussions above.
>
> ---
> Thank you again for your valuable feedback. We hope this rebuttal resolves your concerns and enhances the clarity of our work. We remain open to any further suggestions for improvement.
>
> ---
> [1] Thinking-while-Generating: Interleaving Textual Reasoning throughout Visual Generation, CVPR’26
>
> [2] Vision as a Dialect: Unifying Visual Understanding and Generation via Text-Aligned Representations, NeurIPS’25
>
> [3] Janus-Pro-R1: Advancing Collaborative Visual Comprehension and Generation via Reinforcement Learning, NeurIPS’25

---

### Official Review · Reviewer_AGrv · 2026-03-13

**Soundness:** 3
**Presentation:** 4
**Significance:** 3
**Originality:** 3
**Overall Recommendation:** 5
**Confidence:** 3

**Summary:**

This paper proposes LatentMorph, a framework for integrating latent reasoning into autoregressive text-to-image generation. The key idea is to avoid explicit textual chain-of-thought during generation and instead interleave reasoning in latent space through four components: a condenser for summarizing generation states, a translator and shaper for turning latent reasoning into generation control, and a learned invoker that decides when reasoning should be triggered. Experiments on GenEval, T2I-CompBench, T2I-CompBench++, WISE, and IPV-Txt show strong improvements over the Janus-Pro base model and competitive reasoning-based baselines, with additional gains in efficiency and a fairly extensive set of ablations

**Compliance With Llm Reviewing Policy:**

Affirmed.

**Final Justification:**

My concerns have been resolved.

**Key Questions For Authors:**

See Weaknesses.

**Limitations:**

Yes.

**Strengths And Weaknesses:**

**Strengths**:

The paper tackles an important and timely problem. The method is nontrivial and represents a meaningful combination of latent reasoning and interleaved reasoning for image generation. The experimental results are strong, the work appears substantial, and the ablations help support the contribution of the proposed components. Overall, this is a well-executed exploration of a promising direction and is interesting to the community.

**Weaknesses**:
- The method contains several learned components, which makes the design fairly heavy and may introduce additional training difficulty and complexity.
- Particularly, my main concern is the learned invoker. The paper shows that it outperforms random and fixed invocation, and that its decisions are somewhat aligned with human judgments, but this is not enough to establish that it is the most effective or necessary one for the model. It seems plausible that simpler heuristic strategies, such as entropy- or confidence-based triggering (e.g., [1]), could achieve similar gains with less training complexity. In addition, it would help if there are more illustrative examples to show when the invoker works and how it is necessary.

[1] Chen, Harold Haodong, et al. "Go with Your Gut: Scaling Confidence for Autoregressive Image Generation." arXiv preprint arXiv:2509.26376 (2025).

---

> ### Author Rebuttal · Authors · 2026-03-30
>
> We sincerely thank you for your careful comments and thorough understanding of our paper! We are also encouraged that the reviewers recognized the importance of the problem, our empirical evaluation, and the coherence of the proposed framework.
>
> Your insightful feedback has helped us clarify and refine our contributions. Here we give point-by-point responses to your comments:
>
> ---
> > `Weakness 1`: The method contains several learned components, which makes the design fairly heavy and may introduce additional training difficulty and complexity.
>
> We understand the concern that introducing new modules might seem architecturally heavy and complicate training. However, we respectfully clarify that LatentMorph is intentionally lightweight and highly modular:
>
> - **Minimal Parameter Overhead:** The new components are mathematically minimal (*e.g.*, a MLP-based invoker, 4/8-token condenser), adding **<8.5%** to the base model's parameter count. Moreover, LatentMorph uses **two-stage**, decoupled optimization rather than jointly training all added modules, which substantially reduces training complexity.
>
> - **"Frozen Backbone" Capability:** To further prove this, we train LatentMorph while *strictly freezing the entire Janus-Pro backbone*, updating only our lightweight modules. As shown in **Table A**, this setting still significantly outperforms the vanilla baseline. This provides direct evidence that the gains are not merely due to brute-force backbone tuning.
>
> - **Simple Training Recipe:** Unlike explicit CoT methods [1] requiring expensive step-by-step reasoning data construction, we use only **standard** text-image pairs and preference rewards.
>
>     *Table A: Parameter overhead and frozen backbone ablation*
>     |Setting|Backbone|GenEval|T2I-CompBench|
>     |-|-|-|-|
>     |Vanilla|N/A|0.80|39.21|
>     |SFT|Trainable w/ LoRA|0.79|39.43|
>     |GRPO|Trainable w/ LoRA|0.82|41.61|
>     |**LatentMorph w/ Frozen Backbone**|Frozen|0.94|61.59|
>     |**LatentMorph (Default)**|Trainable w/ LoRA|**0.96**|**64.53**|
>
> ---
> > `Weakness 2`: Necessity of the learned invoker vs. intrinsic confidence signal in [2].
>
> We appreciate your insightful question and the reference to the relevant work on confidence scaling [2]. It is natural to ask if simpler signals could replace our learned invoker. However, we respectfully clarify that [2] and LatentMorph target fundamentally different mechanisms: **sample-level test-time scaling** vs. **step-level in-process intervention**.
>
> - **Different Objectives:** As demonstrated in [2], signals like confidence are highly effective for *test-time scaling*, acting as a lightweight internal signal to **filter and select potential** high-quality *complete samples* without relying on external reward models. In contrast, LatentMorph’s goal is to find the more **precise** *intervention point* (the "when") **within a single ongoing generation stream** to pause and refine the generation trajectory.
>
> - **The "Texture vs. Structure" Bottleneck:** AR models naturally show high entropy when generating valid textures (*e.g.*, grass). A confidence/entropy-only threshold may erroneously triggers redundant interventions here, wasting computation. This issue is less pronounced in [2], because its group-based scaling operates at the sample level for reranking/selection.
>
> - **The Learned Policy:** Our invoker contextualizes uncertainty ($u_i$) with semantic consistency ($c_i$). The RL policy learns to ignore harmless entropy spikes during texture generation, and to intervene primarily when uncertainty coincides with a drop in prompt alignment (*i.e.*, true structural bottlenecks).
>
> To empirically validate this distinction, we further implement confidence-based triggering for our in-process intervention. As shown in **Table B**, our learned policy is necessary to achieve the optimal quality-efficiency balance.
>
> *Table B: Learned invoker vs. confidence-based strategies (T2I-CompBench)*
> |Invocation Strategy|Avg. Invocations/Image|Performance|
> |-|-|-|
> |Random ($p=0.5$)|3.44|59.54|
> |Confidence-based (in [2])|2.56|60.12|
> |**Our Learned Policy**|**1.42**|**64.53**|
>
> ---
> We hope this rebuttal sufficiently addresses your concerns and clarifies the points raised. We are open to further discussion and greatly appreciate your thoughtful feedback.
>
> ---
> [1] Thinking-while-Generating: Interleaving Textual Reasoning throughout Visual Generation, CVPR’26
>
> [2] Go with Your Gut: Scaling Confidence for Autoregressive Image Generation

---

> > ### Author Rebuttal · Reviewer_AGrv · 2026-04-03
> >
> > Thank you for the clarifications and new results. I will maintain my positive judgement.

---

> > > ### Author Response · Authors · 2026-04-03
> > >
> > > **Dear Reviewer AGrv**,
> > >
> > > Thank you very much for your time, your positive feedback, and for maintaining your supportive evaluation of our work.
> > >
> > > Your original questions, particularly regarding the necessity of the learned invoker versus heuristic strategies, were highly constructive and helped us significantly strengthen the empirical validation of our work. We are also deeply encouraged by your recognition of LatentMorph as a **"well-executed exploration" of a timely problem**, and your appreciation for our **"meaningful combination of latent and interleaved reasoning"** for image generation.
> > >
> > > Thank you once again for your rigorous review, your valuable suggestions, and your support of our research.
> > >
> > > Best regards,
> > >
> > > Authors

---

### Decision · Program_Chairs · 2026-04-30

**Decision:**

Accept (regular)

**Comment:**

This submission presents a compelling shift in how we think about "thinking" for T2I. While the current paradigm has been obsessed with making models talk to themselves via explicit text tokens, this work correctly identifies that discrete text is an information bottleneck. By moving reasoning into the continuous latent space, the authors achieve a 25% boost on T2I-CompBench while actually making the model faster—a rare "win-win" in efficiency and quality.

Despite its clear performance gains, the framework is admittedly engineering-heavy, relying on a four-module pipeline (condenser, translator, shaper, and invoker) and a multi-stage training process. However, the rebuttal effectively demonstrated that the training cost is relatively minor (~7 hours) and that the method generalizes across diverse architectures like Tar and LlamaGen. By refining the overclaimed "ineffable semantics" into the more grounded "high-dimensional visual features," the authors have delivered a solid, technically sound contribution. It’s a meaningful step toward more "human-like," event-driven reasoning in vision.